



**Global deposition of total reactive nitrogen oxides from**
**1996 to 2014 constrained with satellite observations of NO$_2$**
**columns**
**Jeffrey A. Geddes[1,2], Randall V. Martin[1,3]**
(1){Department of Physics and Atmospheric Science, Dalhousie University, Halifax, Nova
Scotia, Canada}
(2){Now at: Department of Earth and Environment, Boston University, Boston,
Massachusetts, USA}
(3){Harvard-Smithsonian Center for Astrophysics, Cambridge, Massachusetts, USA}
Correspondence to: J.A. Geddes (jgeddes@bu.edu)
**Abstract**
Reactive nitrogen oxides (NO$_y$) are a major constituent of the nitrogen deposited from the
atmosphere, but observational constraints on their deposition are limited by poor or
nonexistent measurement coverage in many parts of the world. Here we apply NO$_2$
observations from multiple satellite instruments (GOME, SCIAMACHY, and GOME-2) to
constrain the global deposition of NO$_y$ over the last two decades. We accomplish this by
producing top-down estimates of NO$_x$ emissions from inverse modeling of satellite NO$_2$
columns over 1996-2014, and including these emissions in the GEOS-Chem chemical
transport model to simulate chemistry, transport, and deposition of NO$_y$. Our estimates of
long-term mean wet nitrate (NO$_3^-$) deposition are highly consistent with available
measurements in North America, Europe, and East Asia combined (r = 0.83, normalized mean
bias = -7%, N = 136). Likewise, our calculated trends in wet NO$_3^-$ deposition are largely
consistent with the measurements, with 129 of the 136 gridded model-data pairs sharing
overlapping 95% confidence intervals. We find that global mean NO$_y$ deposition over 1996-
2014 is 56.0 Tg N yr$^{-1}$, with a minimum in 2006 of 50.5 Tg N and a maximum in 2012 of 60.8
Tg N. Regional trends are large, with opposing signs in different parts of the world. Over
1996 to 2014, NO$_y$ deposition decreased by up to 60% in eastern North America, doubled in





regions of East Asia, and declined by 20% in parts of Western Europe. About 40% of the
global $NO_y$ deposition occurs over oceans, with deposition to the North Atlantic Ocean
declining and deposition to the northwestern Pacific Ocean increasing. Using the residual
between $NO_x$ emissions and $NO_y$ deposition over specific land regions, we investigate how
$NO_x$ export via atmospheric transport has changed over the last two decades. Net export from
the continental United States decreased substantially, from 2.9 Tg N $yr^{-1}$ in 1996 to 1.5 Tg N
$yr^{-1}$ in 2014. On the other hand, export from China more than tripled between 1996 and 2011
(from 1.0 Tg N $yr^{-1}$ to 3.5 Tg N $yr^{-1}$), before a striking decline to 2.5 Tg N $yr^{-1}$ by 2014. We
find that declines in $NO_x$ export from some Western European countries have counteracted
increases in emissions from neighbouring countries to the east. A sensitivity study indicates
that simulated $NO_y$ deposition is robust to uncertainties in $NH_3$ emissions with a few
exceptions. Our novel long-term study provides timely context on the rapid redistribution of
atmospheric nitrogen transport and subsequent deposition to ecosystems around the world.
**1   Introduction**
The introduction of reactive nitrogen to the environment by anthropogenic activities (e.g.
from fossil fuel combustion and the production of fertilizers for agriculture) has drastically
altered the global nitrogen cycle with consequences throughout the Earth system (Galloway
2004). Reactive nitrogen dominates the chemical production of tropospheric ozone and
contributes to inorganic aerosol formation, with implications for air quality and climate.
Deposition of nitrogen from the atmosphere has been linked to eutrophication and
acidification (Bouwman et al., 2002), reductions in biodiversity (Bobbink et al., 2010;
Hernández et al., 2016; Isbell et al., 2013; De Schrijver et al., 2011), and climate impacts
through coupling with the carbon cycle and greenhouse gas emissions (Liu and Greaver,
2009; Reay et al., 2008; Templer et al., 2012). Despite its global importance, observational
constraints on nitrogen deposition are lacking in many parts of the world due to poor or
nonexistent measurement coverage (Vet et al., 2014).
Atmospheric transport is a dominant process for distributing reactive nitrogen around the
world (Galloway et al., 2008). Some forms of reactive nitrogen can be transported over
distances greater than 1000 km (Neuman et al., 2006; Sanderson et al., 2008), depositing
across national boundaries and continents. For example, the U.S. is estimated to export 30-
40% of its reactive nitrogen emissions (Dentener et al., 2006; Holland et al., 2005; Zhang et





al., 2012), while transport from China accounts for up to 66-92% of total nitrogen deposition
to parts of the northwestern Pacific Ocean (Zhao et al., 2015). Fertilization of the open ocean
due to atmospheric transport and deposition of anthropogenic nitrogen may be a considerable
factor in marine productivity (Duce et al., 2008), prompting important questions about the fate
and impact of deposition far downwind of sources where observations are limited.
Reactive nitrogen oxides ($NO_y \equiv NO + NO_2 + HNO_3 + HONO +$ organic nitrate
molecules + aerosol nitrate) contribute about half of the total nitrogen deposited worldwide
(Dentener et al., 2006). $NO_y$ deposition was estimated to be around 45-50 Tg N yr$^{-1}$ in the
late-1990s and early 2000s, representing a 3-4 fold increase since the pre-industrial era
(Dentener et al., 2006; Kanakidou et al., 2016; Lamarque et al., 2013). A substantial range
exists in the trajectory of global $NO_y$ deposition beyond the year 2000 depending on emission
scenario. Galloway et al. (2004) projected that $NO_y$ deposition could increase by >70% by
2050, while Dentener et al. (2006) projected changes between -25% to +50% by 2030 for
maximum feasible reduction and "pessimistic" scenarios respectively. More recent multi-
model projections by Lamarque et al. (2013) estimate $NO_y$ deposition would change by -16%
to +5% for 2030 and by -48% to -25% for 2100, depending on the Representative
Concentration Pathway (RCP) scenario. This range in projections highlights the need for the
global observational constraints on contemporary changes in nitrogen oxide emissions.
Sources of $NO_x$ ($\equiv NO + NO_2$), whose oxidation is responsible for the formation of other
reactive nitrogen oxides, include fossil fuel combustion, biomass burning, lightning, and
biogenic emission from soil. The magnitude and spatial distribution of $NO_x$ emissions have
changed considerably over the past several decades, corresponding to patterns of human
development and emission control measures. Tropospheric $NO_2$ columns derived from
satellite remote sensing observations have been used extensively to constrain regional and
global $NO_x$ emissions (Streets et al., 2013). This top-down approach complements bottom-up
inventories that are assembled using regionally specific emission factors and fuel combustion
data for various source categories. In particular, satellite $NO_2$ observations can provide insight
into otherwise poorly constrained sources (Beirle et al., 2010; Jaegle et al., 2005; Richter et
al., 2004; Vinken et al., 2014), produce coherent long-term trends (van der A et al., 2008; Lu
et al., 2015; Stavrakou et al., 2008; Zhang et al., 2007), document interannual variability
(Castellanos and Boersma, 2012; Konovalov et al., 2010; Russell et al., 2012), offer
information to evaluate and improve bottom-up inventories at the global scale (Martin, 2003;



Miyazaki et al., 2016), and provide timely emission updates (Lamsal et al., 2011; Mijling et
al., 2013; Souri et al., 2016).

3       Satellite observations of $NO_2$ began with GOME (1995-2003) followed by
SCIAMACHY (2002-2011), and continue today with OMI (2004- ), GOME-2 (2007- ), and
TROPOMI (2017- ), resulting in a record of global atmospheric $NO_2$ abundance over the past
20 years. These observations have been used previously to estimate the deposition of nitrogen
species, either in combination with chemical transport modeling or with empirical approaches
(Cheng et al., 2013; Jia et al., 2016; Lu et al., 2013; Nowlan et al., 2014). For example,
Nowlan et al. (2014) demonstrated how satellite-inferred surface concentrations of $NO_2$ can
be combined with modeling to produce spatially continuous estimates of $NO_2$ dry deposition
fluxes.

12       In this study, we expand on the approach of Nowlan et al. (2014) by using the satellite
observations of $NO_2$ columns to constrain total $NO_y$ deposition, including other oxidized
nitrogen species and wet deposition which contribute substantially to $NO_y$ deposition. We
accomplish this by constraining surface $NO_x$ emissions using the satellite observations of
$NO_2$, and simulating subsequent $NO_y$ deposition with a global chemical transport model.
Given the effective mass balance between $NO_x$ emissions and deposition of reactive nitrogen
oxides, observational constraints on $NO_x$ emissions provide a powerful top-down constraint
on deposition (which to our knowledge has not yet been exploited in this way).

20       We leverage the long-term coverage of GOME, SCIAMACHY, and GOME-2
observations to produce a globally consistent and continuous record of $NO_y$ deposition from
1996 to 2014. We highlight long-term trends in satellite-constrained $NO_y$ deposition around
the world and discuss changes in regional export of $NO_x$. Our satellite-constrained estimates
of $NO_y$ deposition are evaluated using measured wet nitrate ($NO_3^-$) deposition from a variety
of sources worldwide. We also explore the sensitivity of the $NO_y$ deposition estimates to
uncertainties in $NH_3$ emissions.





## 2   Methods

### 2.1   Satellite-based constraints on $NO_y$ deposition

The application of satellite $NO_2$ column observations to constrain $NO_y$ deposition requires a method to propagate the observational information across the entire $NO_y$ system containing species with lifetimes of days or longer. The short $NO_x$ lifetime of hours during daytime satellite observations implies that a direct assimilation for $NO_2$ column abundance would rapidly lose the observational information as the assimilation returns to its unperturbed state well before the next satellite observation days later. We therefore apply satellite $NO_2$ observations to constrain $NO_x$ emissions in a simulation of $NO_y$ deposition so the information propagates across the entire $NO_y$ system.

We calculate top-down surface $NO_x$ emissions from 1996 to 2014 using observations from GOME (1995-2003), SCIAMACHY (2002-2011) and GOME-2 (2007- ). The similar overpass time of these three instruments facilitates their combination to provide consistent long-term coverage (Geddes et al., 2015; Hilboll et al., 2013). Tropospheric $NO_2$ vertical column densities are provided by KNMI at http://www.temis.nl/airpollution/. In all cases, $NO_2$ column densities are retrieved by differential optical absorption spectroscopy using backscattered radiance in the 425-450 nm wavelength range according to the retrieval algorithm documented in Boersma et al. (2004). Boersma et al. (2016) well describe the value of accounting for vertically-resolved instrument sensitivity. We use the averaging kernels provided with the data to replace a priori $NO_2$ vertical profiles with those from GEOS-Chem model following Lamsal et al. (2010). We use daily nadir observations with a cloud radiance fraction of less than 0.5. We minimize errors associated with wintertime retrievals by using a solar zenith angle cut-off of 50°.

We use the GEOS-Chem chemical transport model (www.geos-chem.org) v9-02 to conduct the inversion of satellite observations of $NO_2$ and constrain global $NO_y$ deposition. The simulation is described in Appendix 1. Briefly, GEOS-Chem is driven by assimilated meteorology from the NASA Global Modeling and Assimilation Office and simulates detailed $HO_x$-$NO_x$-VOC-aerosol chemistry (Bey et al., 2001; Park et al., 2004). Removal occurs through wet deposition (Amos et al., 2012; Liu et al., 2001), and dry deposition based on the widely used resistance-in-series formulation (Wesely, 1989). Anthropogenic, biogenic, soil, lightning, and biomass burning emissions are included (see Appendix 1). In the case of



NO$_x$, the bottom-up emissions are used as prior estimates that we then overwrite with the top-
down emissions.

3       We adopt a finite-difference mass balance inversion (Lamsal et al., 2011; Cooper et al.,

*submitted*) for computational expedience given the 19-year period of interest. In two initial
simulations, a perturbation (30%) to the a-priori emissions, *E*, in a grid cell is used to find the
relationship between the a-priori NO$_2$ column, $\Omega$, and the change in the column resulting from
that perturbation:
$$\frac{\Delta E}{E} = \beta \times \frac{\Delta \Omega}{\Omega} \qquad (1)$$
The factor $\beta$ in Equation 1 accounts for non-linear feedbacks between a change in NO$_x$
emissions and NO$_x$ chemistry in a grid cell leading to grid cell NO$_2$ column abundance.
We then use monthly-mean gridded satellite observations, $\Omega_{sat}$, in combination with
monthly $\beta$ values for each grid box to derive new gridded annual emissions, $E_{topdown}$, from the
prior emissions estimates, $E_{prior}$, by rewriting Equation 1 as:
$$E_{topdown} = E_{prior} \cdot [1 - \beta \frac{\Omega_{sat} - \Omega_{prior}}{\Omega_{prior}}] \qquad (2)$$
In all cases, monthly mean simulated NO$_2$ columns are calculated using days with coincident
satellite observations. We calculate annual mean scaling factors from the mean monthly top-
down emissions. The top-down emissions are then used in a final simulation. For locations
without satellite observations, the a-priori emissions are used. The resultant simulation of NO$_y$
deposition is thus constrained by, and consistent with, the satellite NO$_2$ observations (similar
in essence to an assimilation system). We note uncertainty in tropospheric NO$_2$ from lightning
will propagate into the inversion (Travis et al. 2016), but there is no evidence of a significant
trend in lightning NO$_x$ over the long term (Murray et al. 2012). A constant bias is unlikely to
affect the trend analyses presented here.
We derive global mean satellite-constrained NO$_x$ emissions from 1996-2014 of 53.2 $\pm$
3.4 Tg N yr$^{-1}$. Our top-down global NO$_x$ emissions for 2001 of 50.0 Tg N are consistent with
the mean $\pm$ standard deviation from over 20 models used in the Coordinated Model Studies
Activities of the Task Force on Hemispheric Transport of Air Pollution (HTAP) for the same
year of 46.6 $\pm$ 7.8 Tg N (Vet et al. 2014).





**2.2  Measurements of Wet Deposition**
We use a variety of regional and global measurements of wet nitrate ($NO_3^-$) deposition
to evaluate our satellite-constrained simulation from 1996 to 2014.
To evaluate overall global performance we use data compiled by the World Data Centre
for Precipitation Chemistry for two time periods: 2000-2002 and 2005-2007
(http://www.wdcpc.org/). The use of this data ensures optimal and consistent quality
standards across all stations, allowing for evaluation of global performance with careful
consideration of sampling protocols and data completeness (Vet et al. 2014).
To evaluate the long-term means and trends from 1996-2014, we obtain observations of
wet $NO_3^-$ deposition from North America, Europe, and East Asia where continuous
measurements are available throughout most of our study period. Observations come from the
National Atmospheric Deposition Program in the United States (http://nadp.sws.uiuc.edu/,
available 1996-2014), from the  Canadian Air and Precipitation Monitoring Network in
Canada (http://www.ec.gc.ca/rs-mn/, available 1996-2011), from the European Monitoring
and Evaluation Programme in Europe (http://www.emep.int/, available 1996-2014) and from
the Acid Deposition Monitoring Network in East Asia (http://www.eanet.asia, available 2000-
2014). In the US and Canada, wet deposition is measured by wet-only samplers that are
triggered at the onset of precipitation. Measurements in Europe are made by bulk- and wet-
only sampling methods and we used both in this analysis. Measurements across East Asia are
reported as wet-only, although at some stations this may not be accomplished by strictly wet-
only samplers (http://www.eanet.asia/product/manual/prev/techwet.pdf).
For our analysis, we only included stations which had quality controlled annual data for
at least 15 of the 19 years in our study. This left 128 stations across the United States, 14
stations in Canada, 18 stations across Europe, and 14 stations across East Asia. For
comparison with the GEOS-Chem model, if multiple stations are available within a single
grid box we grid all measurements of annual wet deposition to the model horizontal
resolution.
**3  Satellite-Constrained Estimates of $NO_y$ Deposition**
Here we summarize the overall patterns in long-term mean deposition resulting from our
satellite-constrained simulation, followed by a discussion of the long term trends, changes in



regional export, and the sensitivity of the simulated $NO_y$ deposition to potential uncertainties
in $NH_3$ emissions.

### 3.1 Long-term Mean $NO_y$ Deposition

Figure 1 (top) shows our satellite-constrained long-term mean $NO_y$ deposition from
1996 to 2014. We find that 32.2 Tg N $yr^{-1}$ is deposited on average over the continents (57% of
the total), and 23.8 Tg N $yr^{-1}$ is deposited on average over the oceans (43% of the total). This
is similar to the estimate by Galloway et al. (2004) that 46% of modern day $NO_y$ deposition
occurs over the oceans. Critical nitrogen deposition loads for various natural freshwater and
terrestrial ecosystems lie in the range of 5-20 kg N $ha^{-1}$ $yr^{-1}$, depending on the ecosystem, soil
conditions, and land history (World Health Organization, 2000). We estimate that mean
deposition of oxidized nitrogen alone exceeds 5 kg N $ha^{-1}$ $yr^{-1}$ over a land area of
approximately 12.7 x $10^6$ $km^2$ (or ~8% total land area).
In the Northern Hemisphere, high $NO_y$ deposition tends to be associated with regions
that have high anthropogenic $NO_x$ sources. We find mean $NO_y$ deposition in the eastern
United States exceeds 10 kg N $ha^{-1}$ $yr^{-1}$ (maximum = 11.4 kg N $ha^{-1}$ $yr^{-1}$) with elevated
deposition extending into southeastern Canada and hundreds of kilometers into the Atlantic
Ocean. This is similar to the multi-model ensemble results from ACCMIP and HTAP,
predicting between 5-15 kg N $ha^{-1}$ $yr^{-1}$ in this region (Lamarque et al. 2013; Vet et al. 2014).
A prior GEOS-Chem analysis over North America for the years 2006-2008 also predicted
$NO_y$ deposition exceeding 10 kg N $ha^{-1}$ $yr^{-1}$ in the eastern US (Zhang et al. 2012). Elsewhere
in North America, we find high $NO_y$ deposition along the west coast of California (up to 6 kg
N $ha^{-1}$ $yr^{-1}$) and in the vicinity of Mexico City (up to 10 kg N $ha^{-1}$ $yr^{-1}$).
We find mean $NO_y$ deposition is also elevated throughout Europe, with a maximum of
8.5 kg N $ha^{-1}$ $yr^{-1}$ located in northern Italy near the Po Valley region. Again, our long-term
estimate in this region is similar to the ACCMIP and HTAP ensemble means, predicting $NO_y$
deposition in the range of 5-10 kg N $ha^{-1}$ $yr^{-1}$ (Lamarque et al. 2013; Vet et al. 2014). The
elevated deposition here is also spatially consistent with the results from Holland et al. (2005).
We find high deposition extending into western Russia with a hotpot in the vicinity of
Moscow approaching 5 kg N $ha^{-1}$ $yr^{-1}$. Our observation-constrained estimate also has isolated
regions of high deposition in the Middle East (around 4-5 kg N $ha^{-1}$ $yr^{-1}$ in the vicinity of
Tehran and around the Persian Gulf).





We find that the highest mean deposition in the world occurs in China, exceeding 10 kg
N ha$^{-1}$ yr$^{-1}$ in many regions. High deposition extends into the mid-latitude western Pacific
Ocean off the coast of East Asia. NO$_y$ deposition in the ACCMIP and HTAP ensemble means
also exceeds 10 kg N ha$^{-1}$ yr$^{-1}$ throughout eastern China. We find the highest long-term mean
deposition (with a maximum close to 20 kg N ha$^{-1}$ yr$^{-1}$) occurs in the south, around the Pearl
River Delta and in the vicinity of Guangzhou, although deposition is also high in the regions
just west of Beijing and Shanghai.
In the Southern Hemisphere, high NO$_y$ deposition is associated with biomass burning
and soil NO$_x$ sources, in addition to anthropogenic sources. For example, we find NO$_y$
deposition is between 3 to 5 kg N ha$^{-1}$ yr$^{-1}$ in central and southern Brazil, and in the tropical
rainforests and moist savannahs of Africa. Our estimates in these biomass burning and soil
NO$_x$ dominated regions are also generally consistent with the ACCMIP and HTAP ensemble
estimates (2-5 kg N ha$^{-1}$ yr$^{-1}$). We find NO$_y$ deposition up to 10 kg N ha$^{-1}$ yr$^{-1}$ in the vicinity of
Sao Paulo and Rio de Janeiro, and in the vicinity of Johannesburg and the industrialized
Mpumalanga Highveld of South Africa (all dominated by anthropogenic NO$_x$ emissions). Our
constrained simulation also identifies hotspots of deposition in the vicinity of Melbourne and
Sydney, Australia (~4 kg N ha$^{-1}$ yr$^{-1}$).
Figure 1 (bottom) shows the simulated long-term ratio of dry NO$_y$ deposition to total
(wet + dry) NO$_y$ deposition. Globally, dry and wet deposition contribute roughly equally to
total NO$_y$ deposition (52% and 48% respectively). Dry deposition usually accounts for more
than 50% of the total over the continents and directly off shore whereas wet deposition
dominates over the remote oceans. In the generally arid regions of the world (e.g.
southwestern US, the Sahara Desert, the Arabian Peninsula, and the Gobi Desert) dry
deposition accounts for ~85% or more of the total deposition. Elsewhere, dry deposition
fractions tend to be highest (>60%) nearest to major surface NO$_x$ sources (e.g. eastern US,
Western Europe, and near other major urban centres around the world in addition to the soil
and biomass-burning dominated source regions in South America and Africa). HNO$_3$
typically makes the dominant contribution to dry NO$_y$ deposition, although NO$_2$ and HNO$_3$
can make almost equal contributions in certain high-NO$_x$ environments. Isoprene nitrates and
peroxyaxetyl nitrates comprise ~10-30% of dry NO$_y$ deposition in some densely forested and
high latitude environments respectively.



We evaluate our estimates of $NO_y$ deposition with measured wet $NO_3^-$ from several
sources. Figure 2 shows measurements of annual wet $NO_3^-$ deposition from the World Data
Centre for Precipitation Chemistry, available for two time periods: 2000-2002 (N = 470) and
2005-2007 (N = 484). In both we see the patterns of elevated deposition in eastern North
America, Western Europe, and parts of South and East Asia, with lower deposition in western
North America, across high latitudes in the Northern Hemisphere, and in the available
observations in Africa. High deposition in the Southern Hemisphere is observed between Sao
Paulo and Rio de Janeiro, and just southeast of Johannesburg. Figure 2 also shows the wet
$NO_3^-$ deposition from our constrained simulation during the same two time periods (2000-
2002 and 2005-2007), which exhibits similar patterns found in total $NO_y$ deposition (Figure

11     1).

We find a high degree of consistency between our estimate and the observations for
both the 2000-2002 (N = 306 model-data pairs) and 2005-2007 (N = 310 model-data pairs).
Normalized mean bias (NMB) is -14% and -16% respectively. The vast majority of pairs (>
80%) agree to within 50% of each other. Figure 3 shows scatter plots for specific subsets of
the global data. The agreement for both time periods is strongest over North America (r =
0.92 for both 2000-2002 and 2005-2007, NMB = +1.0% and -5.0% respectively). Robust
model agreement with wet nitrate deposition observations over densely monitored North
America is characteristic of other global studies (Dentener et al. 2006; Lamarque et al. 2013;
Vet et al. 2014). Our agreement is also good in Europe (r = 0.69 and 0.66, and NMB = -31.0%
and -29.8% respectively). The weaker correlation and low bias in this region is likewise
characteristic of global studies, although our spatial correlation (r=0.66-0.69) is on the high
end of previously reported multi-model ensembles (r ~ 0.4-0.6, Dentener et al. 2006;
Lamarque et al. 2013; Vet et al. 2014). The negative bias over Europe compared to North
America has previously been attributed to poor modeling of precipitation, and/or spatial
representativeness of the measurements compared to model resolution. Throughout the rest of
the world (encompassing observations mostly over Asia, but also over eastern Russia, and
some locations in the Southern Hemisphere) the combined spatial coverage of the
observations is very low (N = 53). Normalized mean bias in these estimates is also high
compared to North America (NMB = -19.5% and -17.8% for 2000-2002 and 2005-2007
respectively), and our spatial correlation with the measurements is weak (r = 0.35 and 0.42
respectively). We find that our poor agreement here is disproportionately driven by the two
observations that also have the highest measured deposition in the world: near Port Blair on




the South Andaman Island in the Bay of Bengal, and in the Arunachal Pradesh state in
northeastern India. Agreement is considerably better with the rest of the data (r = 0.78 and
0.72, NMB = +0.01% and -0.01% for 2000-2002 and 2005-2007 respectively). Excluding
these two points substantially improves the global agreement as well (from r = 0.57 to 0.75
and r = 0.59 to 0.75 respectively). Site representativeness, precipitation errors, or uncertainty
in our satellite-constrained $NO_x$ emissions may explain the discrepancy at these two specific
sites.

8       In addition to global data for 2000-2002 and 2005-2007 from the World Data Centre for

Precipitation Chemistry, we also evaluate our estimates of $NO_y$ deposition over the long-term
(1996-2014) using continuous observations provided by regional networks. Figure 4 shows
measured wet $NO_3^-$ deposition over North America, Europe, and East Asia for locations where
at least 15 years of quality-controlled annual data are available. These long-term mean
observations demonstrate many of the same spatial patterns as the time slices from 2000-2002
and 2005-2007. In North America, a relatively smooth gradient is observed from low
deposition in the west to high deposition at sites in the east. In Europe, the highest measured
long-term mean wet $NO_3^-$ deposition occurs at a coastal site in southern Norway, at a site just
east of Copenhagen, and at locations in northern Italy and in Switzerland. At higher latitude
sites, deposition is lower. Across the eastern Asia network, the measurements show highest
deposition at sites in Southeast Asia (e.g. at a location between Kuala Lumpur and Singapore,
and another in the vicinity of Jakarta) and in Japan. The lowest long-term mean deposition
occurs at high latitude sites along the border of Russia and Mongolia, while moderate to high
deposition is measured on the coast of eastern China.

23       In general, our satellite-constrained estimate reflects the spatial variability that is seen in

the measurements. Globally, the correlation between measured $NO_3^-$ deposition and our
estimated wet $NO_3^-$ deposition is excellent (r = 0.83, NMB = -7.7%, N = 136 gridded model-
data pairs). The vast majority of pairs (> 85%) agree to within 50% of each other. For the
individual regions, normalized mean bias in our estimate is smallest over North America
(NMB = +2.4%), and higher over Europe and East Asia (NMB = -32% and -25%
respectively). The spatial correlation over each region is strong (r = 0.89, r = 0.87, and r =
0.69 for North America, Europe, and East Asia respectively), but sample sizes over Europe (N
= 16) and East Asia (N = 11) are small so we emphasize caution in the interpretation of the
statistics for these two regions. The lack of continuous measurement coverage even in parts of





the world with routine network observations highlights the imperative of using other novel observational constraints on deposition (such as the global satellite observations of $NO_2$ used here).

## 3.2 Trends in Global $NO_y$ Deposition from 1996 to 2014

Our long-term satellite-constrained estimate of $NO_y$ deposition facilitates a unique and up-to-date investigation of the changes in $NO_y$ deposition around the world. We calculate linear trends in annual $NO_y$ deposition using the nonparametric Sen's method (Sen, 1968), and test for significance with the nonparametric Mann-Kendall method (Kendall, 1975; Mann, 1945). We treat increasing or decreasing trends as significant if $p < 0.01$. Given that this is a test for linear trends, regions where shorter-term trends in deposition may have changed signs over the period of study could result in erroneous or insignificant trends. Below we discuss particular regions where this is the case.

Figure 5 shows the long-term annual and seasonal trends calculated from our satellite-constrained estimate of total $NO_y$ deposition across 1996-2014 (hatching indicates statistical significance). Figure 6 highlights timeseries of total $NO_y$ deposition over three specific regions covering parts of North America, Western Europe, and East Asia (as outlined in dashed boxes in the top panel of Fig. 5).

Substantial decreases are seen throughout North America extending over the Atlantic Ocean to remote regions. The timeseries for this region (Fig. 6, left) shows that $NO_y$ deposition decreased by almost 40% from 6.4 Tg N $yr^{-1}$ in 1996-1998 to 3.9 Tg N $yr^{-1}$ in 2012-2014. The steepest local decline in the world appears over the Ohio River Valley area, with a maximum near Pittsburgh where $NO_y$ deposition decreased by -0.6 kg N $ha^{-1}$ $yr^{-2}$. $NO_y$ deposition near Pittsburgh decreased from consistently exceeding 15 kg N $ha^{-1}$ $yr^{-1}$ during 1996-2000, to below 6 kg N $ha^{-1}$ $yr^{-1}$ by 2014. The strong decrease in the northeast is consistent with other long-term observational studies for the US (Sickles II and Shadwick, 2007, 2015). Studies of US $NO_x$ emissions derived from satellite observations have also highlighted the remarkable success of emission controls (Duncan et al., 2013; Russell et al., 2012). Our constrained estimate has the steepest declines during the summer (Fig 5, JJA), restricted tightly to the source regions. This also agrees with long-term observations showing the strongest reductions in the summer (Sickles II and Shadwick 2015), consistent with the shorter lifetime of $NO_x$ and efficient dry deposition of $NO_y$ over the forested eastern US. We




find significant decreases far downwind over the Atlantic Ocean during the other months,
when $NO_y$ can be transported farther. The steep change in $NO_y$ deposition in the eastern US
over the last 20 years may have important consequences on tree mortality rates in the region,
which have been demonstrated to be very sensitive to $NO_3^-$ deposition in the range of 5-15 kg
N ha$^{-1}$ yr$^{-1}$ (Dietze and Moorcroft, 2011). The steeply decreasing trends across the US in our
satellite-derived $NO_y$ also support the increasing dominance of reduced nitrogen in total
nitrogen deposition evidenced by observations (Li et al., 2016) and model predictions (Ellis et
al., 2013).

9          We find a small but statistically significant positive trend in $NO_y$ deposition (+0.06 kg

N ha$^{-1}$ yr$^{-2}$) in northern Alberta, Canada, dominated by the trend in JJA. The region is
downwind of development in the Canadian oil sands, which has seen notable changes in $NO_2$
column abundance as observed from space (McLinden et al., 2012). We estimate that
deposition of $NO_y$ in this area was at a maximum of 3.4 kg N ha$^{-1}$ yr$^{-1}$ in 2011 (up from 1.3 kg
N ha$^{-1}$ yr$^{-1}$ in 1996-1997), and has since declined to 1.6 kg N ha$^{-1}$ yr$^{-1}$ by 2014. Elsewhere in
Canada we estimate that $NO_y$ deposition has decreased in the south and east parts of the
country, consistent with observational analyses (Zbieranowski and Aherne, 2011).

17          Declines in $NO_y$ deposition are also found across Europe, but statistical significance

tends to be limited to western continental Europe and the United Kingdom (while changes in
the south, north, and eastern countries tend to be insignificant). According to the timeseries
for this region (Fig. 6, middle), $NO_y$ deposition decreased by about 15% (from 2.5 Tg N yr$^{-1}$
in 1996-1998 to 2.1 Tg N yr$^{-1}$ in 2012-2014). We find the steepest local trends (-0.1 kg N ha$^{-2}$
yr$^{-1}$) in eastern Germany and southern UK, where $NO_y$ deposition in 2012-2014 decreased by
20% compared to 1996-1998. Previous satellite constraints on $NO_x$ emissions established that
$NO_x$ emissions in France, Germany, Great Britain, and Poland have declined since 1996 while
emissions in Greece, Italy, Spain, and the Ukraine for example have either stayed constant or
increased (Konovalov et al., 2008). The local variability in emission trends leads to notable
transboundary impacts. For example, our simulation predicts no net trend in $NO_y$ deposition
over the Ukraine; but we find this is a result of opposing trends in dry (increasing) and wet
(decreasing) deposition. This would be explained by increasing local emissions but decreasing
transport from upwind. Similarly, we find significant increases in dry deposition in parts of
western Russia but no significant trend in wet deposition.





Large increases in $NO_y$ deposition are found throughout Asia, concentrated especially in
eastern China and parts of Southeast Asia. Figure 6 shows the timeseries of wet and dry $NO_y$
deposition within the rectangular region outlined in Figure 5 that encompasses eastern China
and part of the adjacent ocean. We find that $NO_y$ deposition in the region increased by 65%
from 5.2 Tg N yr$^{-1}$ in 1996-1998 to 8.6 Tg N yr$^{-1}$ in 2012-2014. The timeseries also shows
that $NO_y$ deposition decreased after peaking around 9.3 Tg N yr$^{-1}$ in 2011-2012. We find that
the steepest increasing local trends in the world appear in eastern China, and in the Pearl
River Delta region (up to +0.6 kg N ha$^{-1}$ yr$^{-2}$). In fact, deposition in the Pearl River Delta
region is the highest in the world for most of our record, exceeding 20 kg N ha$^{-1}$ yr$^{-1}$ every
year from 2003-2014 (almost doubling from just over 11 kg N ha$^{-1}$ yr$^{-1}$ in 1996). The trends in
deposition over China are largest in summer when the $NO_x$ lifetime is short, with more
obvious indications of increasing $NO_y$ transport/export in the spring months (Fig 5, MAM).
The substantial increase in $NO_x$ emissions throughout East Asia has been inferred from
satellite instruments in several previous studies (Mijling et al., 2013; Richter et al., 2005).
The timing and extent of the reversal in $NO_y$ deposition that we see is also consistent with
observed $NO_2$ columns over eastern China derived from OMI (de Foy et al., 2016; Krotkov et
al., 2016). The ability of our satellite-constrained $NO_y$ deposition estimate to capture this
sudden dramatic decrease over China, in contrast with previous projections (e.g. the RCP2.6,
RCP4.5, and RCP8.5 projections to 2030, Lamaque et al. 2013), emphasizes an attribute of
the satellite constraint.
Small statistically significant decreasing trends are found over the biomass-burning
dominated source regions of Africa. The decrease in $NO_y$ deposition of about ~3% yr$^{-1}$
relative to the long-term mean in Northern Africa is consistent with the most recent GFED4
inventory from 1997 to 2014 (http://www.globalfiredata.org/), which has fire $NO_x$ emissions
decreasing at a rate of about 3% yr$^{-1}$ in this region. In contrast, we also estimate a similar
decrease in Southern Africa that is not represented in the recent GFED4 emission timeseries.
A reduction in $NO_2$ column abundance in this region (observed by GOME and
SCIAMACHY) was also reported by van der A. (2008). They postulate that this decline could
be a result of deforestation leading to less biomass burning, but changing $NO_x$ emission
factors from biomass burning could also potentially explain the trend.
Despite the large regional trends described above, we find that global deposition
changed very little between 1996-1998 (56.1 Tg N yr$^{-1}$) and 2012-2014 (58.5 Tg N yr$^{-1}$) due





to the opposing changes in different regions. Total $NO_y$ deposition was lowest in 2006 (50.5
Tg N), and peaked at 60.8 Tg N in 2012. Since then, it appears that global $NO_y$ deposition
may be on the decline. Future observations in the coming years will be needed to establish
whether this most recent decline is robust or temporary.

5       Figure 7 shows the calculated long-term trends in the measured wet $NO_3^-$ deposition for

locations across North America, Europe, and East Asia where at least 15 years of quality-
controlled annual data are available (the coverage of these observation is the same as in
Figure 4). The observations over North America show the gradient in trend from negligible in
the west to steeply and significantly decreasing in northeastern US and southeastern Canada.
The steepest observed statistically significant trend (-0.18 kg N $ha^{-1}$ $yr^{-2}$) occurs east of
Detroit in southwestern Ontario, Canada. In Europe, only one of the gridded observations has
a statistically significant trend (-0.07 kg N $ha^{-1}$ $yr^{-2}$), located near the border of Denmark and
Germany. The other locations in Europe show statistically insignificant trends over the long-
term. In East Asia, we also we find that most of the stations record statistically negligible
trends over the long term (only two of the 11 gridded observations have significant trends).
The steepest observed trend in this region (+0.39 kg N $ha^{-1}$ $yr^{-2}$) is found near Kuala Lampur
and is statistically significant.

18       We compare the long-term trends in these measurements with our satellite-constrained

trends in wet $NO_3^-$ deposition. We find a similar spatial gradient in North America, and the
same magnitude of declines through the northeast US and southern Ontario (-0.12 to -0.16 kg
N $ha^{-1}$ $yr^{-2}$). Over Europe, our estimates have low statistical significance in the trends
throughout much of this region, consistent with the observations. Where we do see statistical
significance (northern United Kingdom, southern Denmark, and in some central/eastern
European countries), observations are not available over the long-term for evaluation. In East
Asia, our satellite-constrained estimated trends show statistically significant increases
throughout much of the region (in contrast to most of the available observations). The trend
over Kuala Lampur is significant and positive (+0.24 kg N $ha^{-1}$ $yr^{-2}$) as expected from the
available measurements.

29       We again emphasize the small sample size in Europe (N = 16) and East Asia (N = 11).

Moreover, in many cases trends in one (or both) datasets are small and/or insignificant. For
these reasons, we focus on comparing the confidence intervals of the measured and satellite-
constrained trends. We find that for 129 of the 136 gridded pairs (> 90% of the data), the 95%



confidence intervals overlap; of the pairs for which the intervals do not overlap, 3 (out of 109)
occur in North America, 1 (out of 16) in Europe, and 3 (out of 11) in East Asia. For a large
majority of the data in all three regions we therefore conclude that the satellite-derived trends
are not significantly different from the trends inferred with ground-based measurements.
Continued long-term measurements with better spatial coverage are imperative to better
evaluate long-term estimates of global $NO_y$ deposition especially throughout Europe and East
Asia (but also in other parts of the world where long-term coverage is not available at all).
### 3.3   Changes in Continental Export of NOy

9       $NO_y$ deposition is a transboundary, and even intercontinental, issue (HTAP, 2010). In a

multi-model study, Sanderson et al. (2008) found that between 3-10% of $NO_x$ emissions from
Europe, North America, South Asia, and East Asia are ultimately deposited over foreign
regions. Long range transport events of $NO_2$ alone can be systematically detected by satellite
observations (Zien et al., 2014). Here we extend such studies using our satellite-constrained
long-term estimates of annual $NO_y$ deposition to evaluate how the amount of $NO_x$ exported
from specific regions (i.e. the net balance between emissions and deposition over a land area)
has changed over the last two decades.

17       Decreases in $NO_y$ export over the Atlantic Ocean from North America and increases in

export over the western Pacific Ocean from East Asia are evident in Figure 5. We find that net
export of $NO_x$ from North America via atmospheric transport has decreased by more than
40% (from 2.5 Tg N yr$^{-1}$ in 1996, to 1.4 Tg N yr$^{-1}$ in 2014). In contrast, we find that export of
$NO_x$ from Asia increased by 40% from 3.3 Tg N in 1996, to a maximum of 4.7 Tg N in 2011,
with a subsequent decrease to 3.8 Tg N by 2014. As a result of these opposing trends, total
deposition to the global oceans has changed remarkably little (25.0 Tg N yr$^{-1}$ in 1996-1998
compared to 24.4 Tg N yr$^{-1}$ in 2012-2014), but has experienced substantial regional
redistribution.

26       $NO_y$ export from North America has received considerable attention. Urban plumes

from the eastern US that are transported across the North Atlantic for several days could still
contain 20-50 ppb of reactive nitrogen oxides (Neuman et al. 2006). A recent detailed GEOS-
Chem study of nitrogen deposition over the US estimated that net annual export of $NO_x$ was
around 38% of $NO_x$ emissions (or 2.5 Tg N) for 2006-2008 (Zhang et al. 2012). We estimate
a similar fraction of export from the continental US using our observationally-constrained





simulation (34% ± 2% from 1996-2014). As a result of declining emissions we find that
absolute export from the continental US decreased by 50% from 2.9 Tg N in 1996 to 1.5 Tg N
in 2014. We find declines in $NO_y$ deposition across the Atlantic Ocean, with small though
statistically significant declines as far downwind as southern Greenland. The decreases
downwind of the continent are clearest and most significant in the winter, spring, and fall
(Fig. 5b, c, and e) while the trends are more local in the summer (when the $NO_x$ lifetime is
short and when midlatitude wind speeds are weaker.
We similarly calculate the net imbalance between $NO_x$ emissions and $NO_y$ deposition
over western European countries and find a decrease of almost 40%, from 2.2 Tg N to 1.3 Tg
N. In contrast to the continental US, we also find a notable decrease in the fraction of
emissions that are exported from western European countries (from 50% in 1996-1998 to
40% in 2012-2014). As a result, the decrease in net export is steeper than the decrease in
emissions from the region. As alluded to in Section 3.2, the decrease in $NO_x$ export from
some western European countries has likely compensated for increases in emissions in some
of the central/eastern European countries, and in western Russia, where we find dry
deposition has significantly increased, but wet deposition has decreased or shows no
significant net trend.
Reactive nitrogen transport from Asia has previously been shown to contribute to $O_x$
production across the mid-latitude Pacific reaching as far as the west coast of North America
(Walker et al., 2010; Zhang et al., 2008), and major $NO_x$ transport events from China can be
indirectly observed by $NO_2$ columns (Lee et al., 2014). Our satellite-constrained estimate
predicts that export from China alone (24% ± 4% of emissions) more than tripled from 1.0 Tg
N in 1996 to a maximum of 3.5 Tg N in 2011, then decreased to 2.5 Tg N by 2014. Zhao et al.
(2015) used GEOS-Chem to explore nitrogen deposition to the northwestern Pacific Ocean
off the coast of China from 2008-2010. They estimated total (wet + dry) $NO_y$ deposition of
6.9 kg N ha$^{-1}$ yr$^{-1}$ and 3.1 kg N ha$^{-1}$ yr$^{-1}$ to the Yellow Sea and the South China Sea
respectively. Our simulation predicts that $NO_y$ deposition to the same regions of the Yellow
Sea increased from 5.1 kg N ha$^{-1}$ yr$^{-1}$ in 1996 to 9.5 kg N ha$^{-1}$ yr$^{-1}$ by 2012 and to the South
China Sea from 2.8 kg N ha$^{-1}$ yr$^{-1}$ in 1996 to 4.3 kg N ha$^{-1}$ yr$^{-1}$ in 2011. Subsequent declines in
the following years will hopefully have encouraging implications for nitrogen availability and
the incidence of algal blooms in these regions (Hu et al., 2010).



1   Export of pollution from China has been shown to influence deposition over Japan in

2  particular (e.g. Lin et al., 2008). Using observations of wet nitrate deposition, Morino et al.

3  (2011) report increases throughout Japan from 1989-2008, and attribute this trend largely to

4  transport from China. Likewise, integrated $NO_y$ deposition over Japan increased ($p < 0.01$) in

5  our satellite-constrained estimate. In fact, we find that Japan transitioned from a net

6  "exporter" of $NO_y$ over 1996-2006 (emissions exceeded local deposition by up to 24%) to a

7  net "importer" of $NO_y$ over 2007-2014 (local deposition exceeded emissions by up to 20%).

8  The increase in deposition was dominated by statistically significant increases in wet

9  deposition in some parts of the country. We find the increase over Japan is most uniform

10  during the spring (Fig. 5, MAM), consistent with transport from China being pronounced

11  during the spring season (Tanimoto et al., 2005). Nevertheless, the impacts of local $NO_x$

12  controls can also be important. Dry deposition dominates the decline in annual $NO_y$

13  deposition just west of Tokyo. Declines are seen throughout the southern part of the country

14  during both the summer and fall seasons (Fig. 5, JJA and SON). These results demonstrate the

15  indirect relationship between local emissions and local deposition of $NO_y$ for regions

16  influenced by atmospheric transport, and also show how long-term trends can depend strongly

17  on the season and process (wet or dry deposition).

18  **3.4 Sensitivity of $NO_y$ Deposition to $NH_3$ Emissions**

19    The transport and ultimate deposition of oxidized nitrogen may be tightly coupled with

20  the reduced nitrogen ($NH_x = NH_3 + NH_4^+$) and sulfate systems, due to the formation of

21  $NH_4NO_3$ aerosol that becomes favorable once all $H_2SO_4$ has been neutralized (i.e., if there is

22  "excess" $NH_3$). Examples of the resulting non-linearity between $PM_{2.5}$ concentrations and

23  precursor emissions have been noted in the literature (Banzhaf et al., 2013; Derwent et al.,

24  2009; Fowler et al., 2005). The formation of $NH_4NO_3$ aerosol at the expense of $HNO_3$ with

25  changing excess ammonia could therefore conceivably change the atmospheric lifetime of

26  $NO_y$ at the surface; accumulation mode aerosol may have a dry deposition lifetime of days

27  whereas $HNO_3$ tends to have a dry deposition lifetime of shorter than a day. As a result, the

28  predicted footprint of source impacts is sensitive to $NH_3$ emissions (Lee et al., 2016).

29    Contemporary emissions of $NH_3$ are highly uncertain (Reis et al., 2009), so we perform

30  a sensitivity experiment by perturbing $NH_3$ emissions everywhere by 25% for the year 2012.

31  Predicted $NO_y$ deposition from this simulation is compared to the predicted $NO_y$ deposition in

32  the 2012 simulation where $NH_3$ emissions were not perturbed. Since we have not altered the



emissions of oxidized nitrogen, simple mass balance dictates that increases in deposition over
some regions will be countered by decreases elsewhere. Our perturbation is therefore to be
interpreted as an experiment that tests how accurately the spatial pattern in $NO_y$ deposition at
our model resolution can be predicted, given some uncertainty in $NH_3$ emissions. Given the
horizontal resolution of our simulation (2.5° x 2.0°), we acknowledge that our estimates of the
sensitivity of $NO_y$ deposition to perturbations in $NH_3$ emissions may underestimate the
importance of those interactions at finer spatial scales.
Figure 8 shows the results of this experiment. The sensitivity of $NO_y$ deposition to an
increase in $NH_3$ emissions is positive or negative depending on the region, while net
deposition over the global domain does not change (to within 1-2%). Over the continents, the
sensitivity in total (wet + dry) $NO_y$ deposition to the 25% perturbation in $NH_3$ emissions tends
to be less than ± 5%, with a few exceptions. We find differences in $NO_y$ deposition on the
order of 10% over parts of high-latitude Russia, northwest and central Africa, eastern China,
southern South America, and Australia. However, with the exception of China, these are also
regions where deposition is relatively low. We conclude that for most regions of interest, our
satellite-constrained estimates of $NO_y$ deposition over the continents and their trends will not
be severely impacted by uncertainty in the $NH_3$ inventories.
Notably, the difference exceeds +50% over Myanmar, suggesting that simulated $NO_y$
deposition over this country is extremely sensitive to changes in $NH_3$ emissions. It is clear
from Figure 8 that this results from a high sensitivity in dry deposition (middle panel) instead
of wet deposition (bottom panel). Myanmar has some of the lowest estimated $NH_3$ emissions
in all of South and East Asia (at least an order of magnitude lower than surrounding India,
China, and Thailand), so this sensitivity reflects changes in the upwind emissions and
subsequent transport of $NO_y$. We find the opposite sensitivity in nearby Cambodia, where the
sensitivity of dry $NO_y$ deposition to a 25% perturbation is $NH_3$ emissions is -50%.
Over the oceans, the sensitivity of $NO_y$ deposition to the 25% increase in $NH_3$ emissions
is generally low (< ± 5%), with the expected exceptions in areas that are directly offshore
from major continental source regions. In the North Atlantic Ocean east of Canada and
Greenland, and in the North Pacific Ocean off the coasts of China, Japan and in the South
China Sea, the sensitivity of $NO_y$ deposition is between 5-20%. Our predicted decrease in dry
$NO_y$ deposition to the Yellow Sea given an increase in $NH_3$ emissions is consistent with
previous adjoint analyses showing increased $NO_y$ dry deposition in this region with a decrease





in Asian $NH_3$ emissions (Zhao et al. 2015). Likewise, the sensitivity of deposition to the
Mediterranean Sea is between 10-20%. The differences in $NO_y$ deposition over the oceans
results from sensitivity in both dry and wet deposition (although in the case of the
Mediterranean it is dominated by dry deposition). We conclude that although changes (or
uncertainties) in $NH_3$ emissions can impact the distance of transport and deposition to oceans
downwind of the major $NO_x$ sources, the absolute magnitude of deposition is low where the
sensitivity of $NO_y$ deposition to $NH_3$ is relatively high.
**4   Conclusion**

10        $NO_y$ deposition represents about half of the total reactive nitrogen deposited to Earth's

surface. Even in the US where nitrogen oxide emissions have decreased substantially,
constituents of $NO_y$ remain major contributors to the nitrogen deposited in areas of concern
(Lee et al. 2016; Li et al. 2016). We applied $NO_2$ observations from multiple satellites over
1996-2014 together with the GEOS-Chem chemical transport model to estimate long-term
changes to reactive nitrogen oxide deposition around the world. Given the effective global
mass balance between $NO_x$ emissions and deposition of reactive nitrogen oxides, we show
that satellite constraints on $NO_x$ emissions can provide a powerful top-down constraint on
deposition in order to evaluate long-term changes worldwide. Observations from the GOME,
SCIAMACHY, and GOME-2 satellite instruments have provided continuous global coverage
over the last 20 years, allowing observational constraints on $NO_y$ deposition that enhance the
poor spatial coverage of ground-based deposition measurements.

22        We find substantial variability in regional trends of $NO_y$ deposition. $NO_y$ deposition

declined most steeply throughout the northeastern United States by up to -0.6 kg N $ha^{-1}$ $yr^{-2}$,
but has also decreased significantly throughout most of the country and in southern Canada.
In Europe, statistically significant declines of up to -0.1 kg N $ha^{-1}$ $yr^{-2}$ are seen over some
western countries. On the other hand, $NO_y$ deposition has increased substantially throughout
East Asia, exceeding +0.6 kg N $ha^{-1}$ $yr^{-2}$ in some parts. Since reductions in deposition over
some regions were counteracted by increases in others, global $NO_y$ deposition did not change
considerably over the long term. However, we find that global $NO_y$ deposition could now be
on the decline overall, since deposition in Asia peaked around 2010-2012. The ability to
resolve the striking recent decline in NOy deposition in China (despite prior projections of



increasing $NO_x$ emissions) demonstrates one of the attributes of using a satellite-based
constraint. Future observations will be important in evaluating whether this trend persists.
We find that changes over the last two decades in the export of reactive nitrogen oxides
via atmospheric transport have impacted countries downwind of source regions. Export from
North America has decreased by at least 40%, while export from Asia has increased by the
same relative amount. We find evidence that decreases in $NO_x$ export from some western
European countries have counteracted increases in local emissions from some eastern/central
European countries, resulting in negligible net change in $NO_y$ deposition over the long term.
Likewise, Japan is highly sensitive to changes in export from China, but this depends strongly
on the season and whether wet and dry deposition are both considered. While uncertainty in
$NH_3$ emissions can impact the footprint of $NO_y$ export and deposition, we show that this
sensitivity is small in most regions of concern.
Direct measurements of deposition are sparse, inhibiting evaluation. This is especially
challenging for global simulations, where individual measurements may not necessarily be
regionally representative. Nevertheless, we find that for the vast majority of locations our
satellite-derived trends are largely consistent with the observed trends. Expanded coverage of
ground-based observations over the long-term is needed to more comprehensively evaluate
long-term estimates of global $NO_y$ deposition. This need also motivates the value of using
alternative observational constraints like the satellite $NO_2$ columns as presented here.
Forthcoming satellite observations of $NO_2$ at higher spatial resolution (e.g. TROPOMI
(Veefkind et al., 2012)) and with diurnally varying observations (e.g. TEMPO (Zoogman et
al., 2016), Sentinel-4, and GEMS) will offer increasingly robust constrains on $NO_x$ emissions
that affect $NO_y$ deposition. Satellite observations of $NH_3$ (e.g. Van Damme et al., 2014) may
offer additional opportunities to constrain the reactive nitrogen budget. Higher resolution
global modeling will also be an important development to accurately account for non-linear
$NO_2$ losses in global emission inversions (Valin et al., 2011).
Our satellite-constrained estimates of $NO_y$ document interannual changes over the past
two decades worldwide. We expect that this information will be useful in future research into
the impacts of nitrogen deposition to important biodiversity hotspots, in regions dealing with
excessive nitrogen inputs leading to algal blooms, or estimating the changing impacts of
nitrogen deposition on global carbon uptake.



## Appendix 1

We simulate atmospheric chemistry from 1996 to 2014 using the GEOS-Chem chemical transport model (www.geos-chem.org) v9-02. Our simulations are driven with the MERRA meteorological product at a global horizontal resolution of 2.5° x 2.0° and 47 vertical layers. GEOS-Chem includes detailed $HO_x$-$NO_x$-VOC-$O_3$-aerosol chemistry (Bey et al. 2001; Park et al. 2004), with isoprene chemistry following Paulot et al. (2009a, 2009b) and gas-aerosol partitioning for the sulfate-nitrate-ammonium system calculated according to the ISORROPIA II equilibrium model (Fountoukis and Nenes, 2007). Gas-aerosol phase coupling occurs via $N_2O_5$ uptake (Evans and Jacob, 2005) and $HO_2$ uptake (Mao et al., 2013) in addition to other heterogeneous chemistry (Jacob, 2000) and aerosol effects on photolysis frequencies (Martin et al., 2003). Our simulations use timesteps of 15 minutes for transport and convection, and 30 minutes for emissions and chemistry.

Removal by wet deposition occurs through scavenging in moist convective updrafts, as well as in-cloud and below-cloud scavenging during large-scale precipitation for water-soluble aerosol and gases (Amos et al., 2012; Liu et al., 2001). Removal by dry deposition is calculated based on the widely used resistance-in-series formulation from Wesely (1989), described for GEOS-Chem in Wang et al. (1998) and Zhang et al. (2001) for aerosol. Dry deposition of $NO_y$ over the United States was recently explored and evaluated in detail by Zhang et al. (2012).

Anthropogenic emissions are prescribed by the NEI 2005 inventory for the United States (http://www.epa.gov/ttnchie1/trends/), the CAC inventory for Canada (http://www.ec.gc.ca/pdb/cac/), the BRAVO inventory for Mexico (Kuhns et al., 2005), the EMEP inventory for Europe (http://www.emep.int/), and Zhang et al. (2009) for China and Southeast Asia. Elsewhere, the EDGAR v3 emission inventory is used for anthropogenic $NO_x$, CO, and $SO_x$ (Olivier et al., 2005), the GEIA inventory for $NH_3$ (Bouwman et al., 1997), and the RETRO inventory for VOCs (Hu et al., 2015). Aircraft emissions are from the AEIC inventory (Stettler et al., 2011). Scale factors based on energy statistics following van Donkelaar et al. (2008) are used to scale $NO_x$, CO and $SO_x$ emissions between 1996 and 2010 for years when the emissions are unavailable from the inventory. For other species and for emissions beyond 2010, the closest available year is used. Biogenic VOC emissions are calculated using the MEGAN model (Guenther et al., 2006). Biomass burning emissions are according to the GFED3 inventory (Mu et al., 2011). Soil $NO_x$ is calculated using the



Berkeley-Dalhousie Parameterization (Hudman et al., 2012). Lightning NOx is implemented
according to Murray et al. (2012). These a-priori surface $NO_x$ emissions are overwritten by
our satellite-derived top-down estimates in the assessment of $NO_y$ deposition.
**Acknowledgements**
This work was supported by NSERC and Environment and Climate Change Canada. We
acknowledge the free use of tropospheric $NO_2$ column data from the GOME, SCIAMACHY,
and GOME-2 sensors from www.temis.nl. We further acknowledge the NADP, CAPMoN,
EMEP and EANET regional monitoring networks, and the World Data Centre for
Precipitation Chemistry for access to wet deposition data.





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



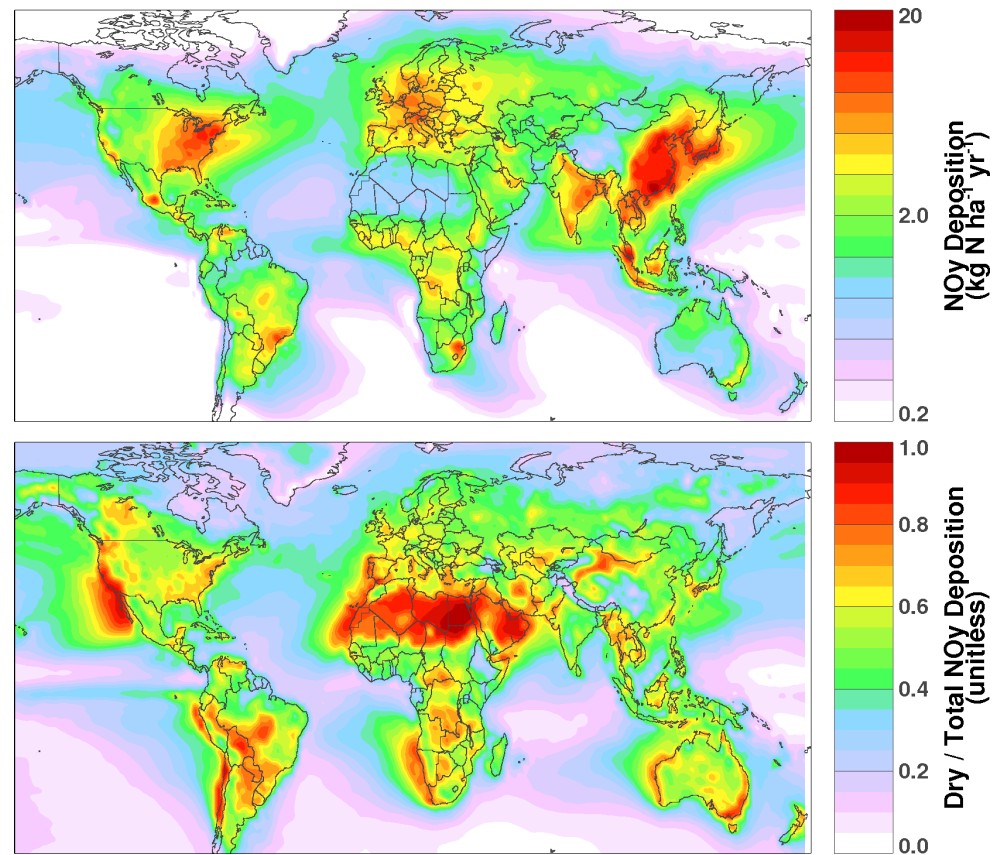

Figure 1: Long-term (1996-2014) mean $NO_y$ deposition derived from the GEOS-Chem
simulation constrained by satellite observations of $NO_2$ columns from the GOME,
SCIAMACHY, and GOME-2 instruments (top). Mean ratio of simulated dry $NO_y$ deposition
to total $NO_y$ deposition (bottom).



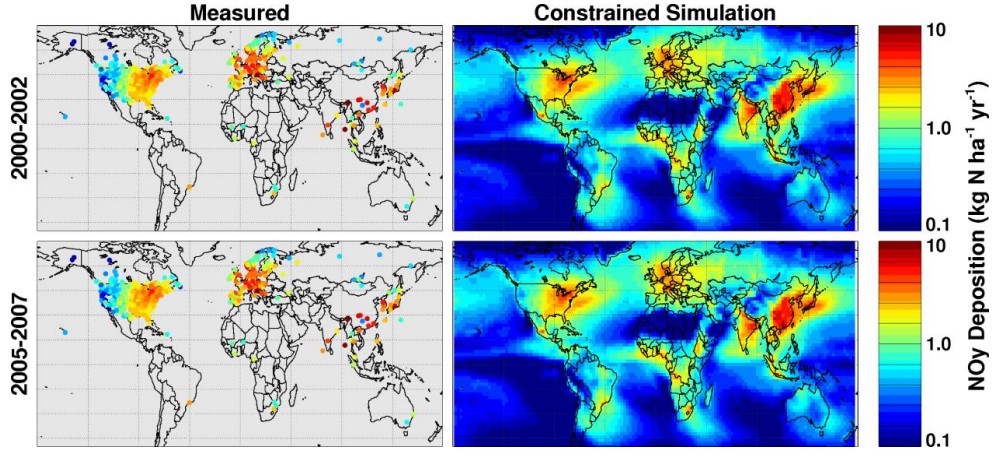

2 Figure 2: Annual wet NO₃- deposition from measurements available through the World Data

3 Centre for Precipitation Chemistry, and from the GEOS-Chem simulation constrained with

4 satellite observations of NO₂. Two time periods are represented: 2000-2002 and 2005-2007.





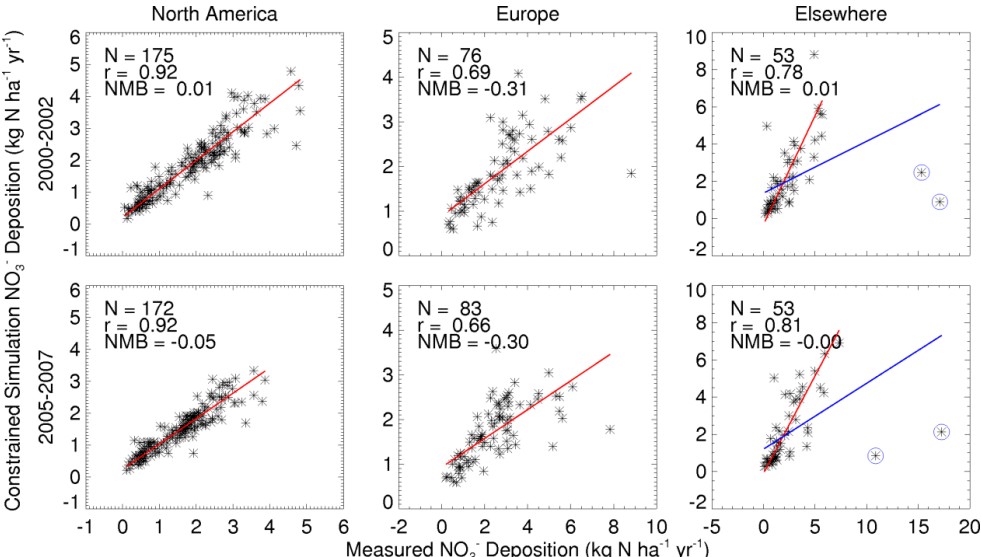

Figure 3: Scatter plot of the satellite-constrained simulated wet $NO_3^-$ deposition vs.
measurements available through the World Data Centre for Precipitation Chemistry for
specific subsets of the data. The red lines show the result of a reduced major axis linear
regression. In the right column, the blue line shows the fit across all data and the red line
shows the fit excluding the two circled data points that are discussed in the text (reported
statistics refer to the red line fit).





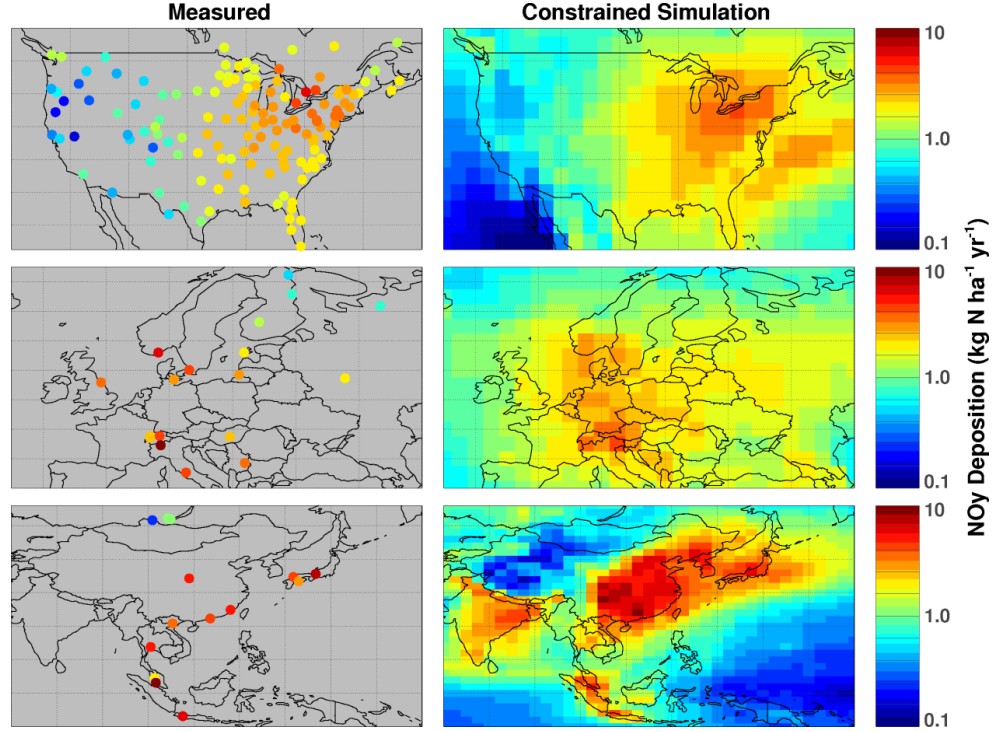

2    Figure 4: Long-term (1996-2014) wet $NO_3^-$ deposition from available regional network

3    measurements (top: NADP and CAPMON; middle: EMEP; bottom: EANet), and from the

4    GEOS-Chem simulation constrained with satellite observations of $NO_2$.



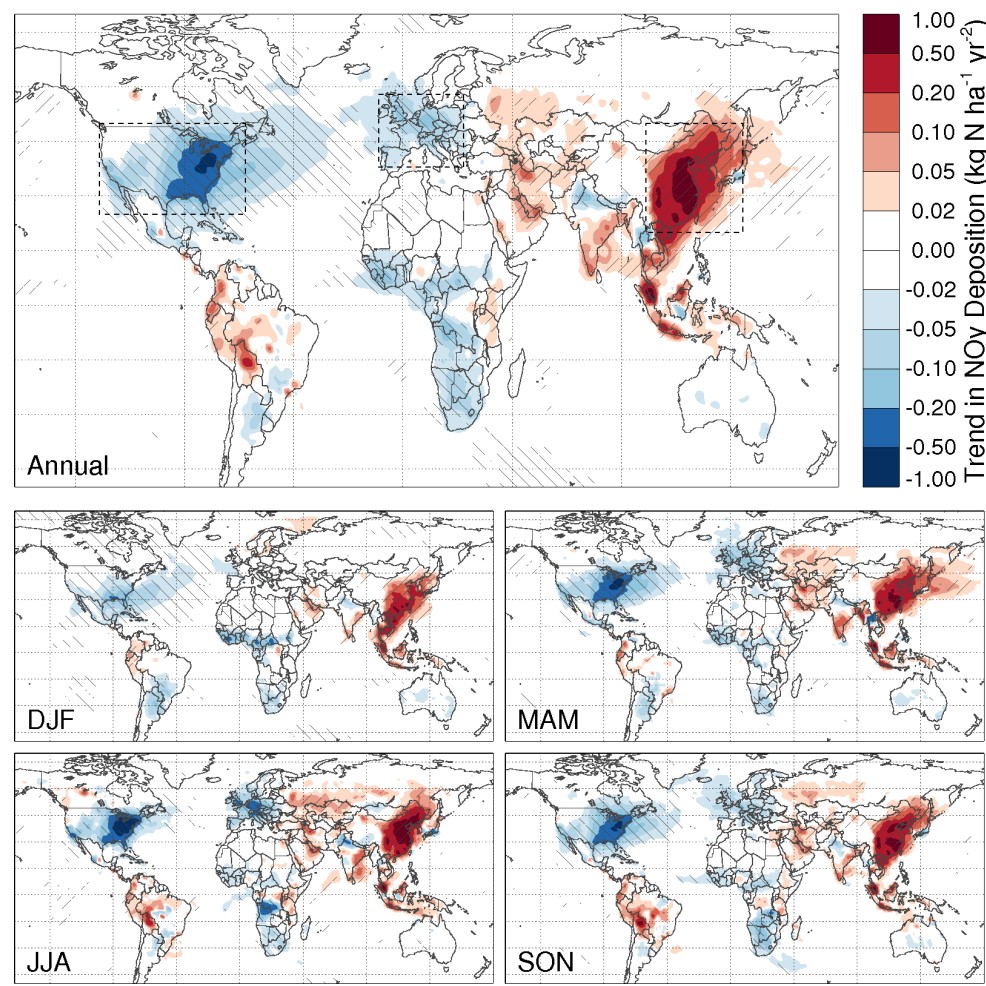

2    Figure 5. Long-term trend (1996-2014) in the satellite-constrained simulation of $NO_y$
deposition. (A) Annual mean; (B) December-January-February; (C) March-April-May; (D)
June-July-August; (E) September-October-November. Diagonal hatching represents trend
significance ($p < 0.01$). Hatching from top-left to bottom-right indicates a decreasing trend;
hatching from bottom-left to top-right indicates an increasing trend.





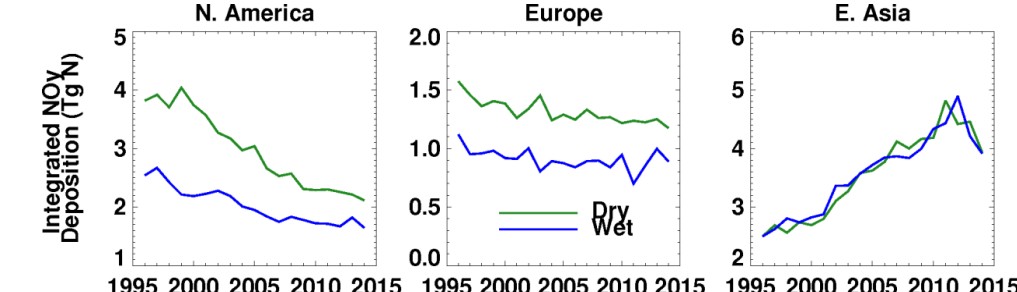

2    Figure 6: Timeseries of annually integrated dry and wet NOy deposition over specific regions

3    (North America, Europe, and East Asia) as defined by the dashed rectangles in Figure 5.





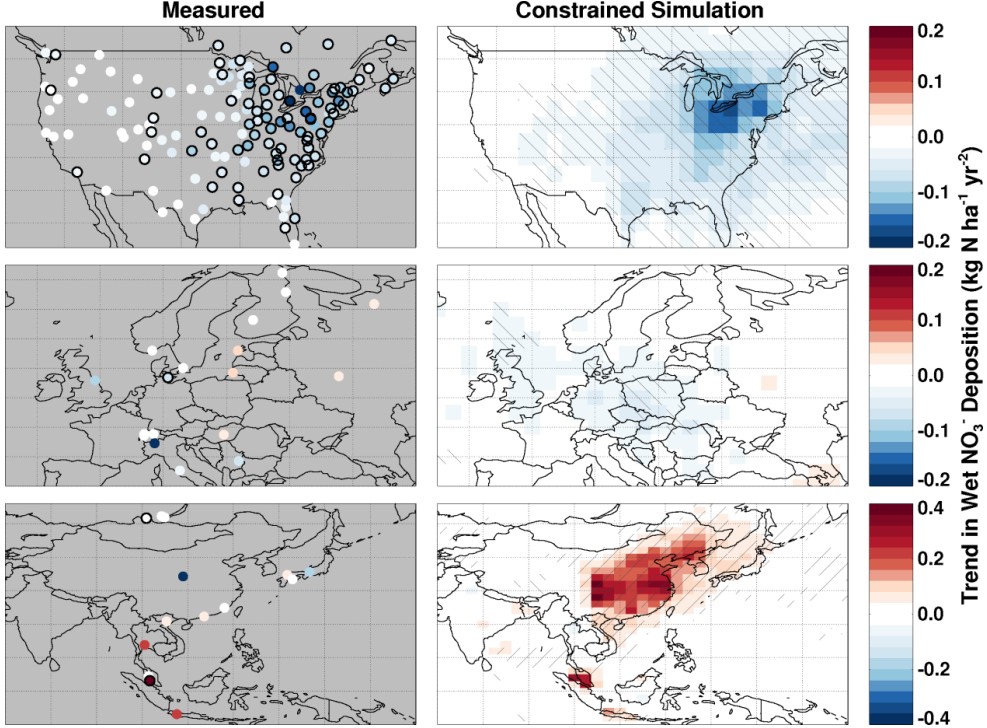

Figure 7: Long-term (1996-2014) trends in wet $NO_3^-$ deposition from available regional network measurements (as in Figure 4), and from the GEOS-Chem simulation constrained by satellite observations of $NO_2$. Closed circles around the measurements indicate significant trends ($p < 0.01$); hatching indicates statistical significance in the simulation.





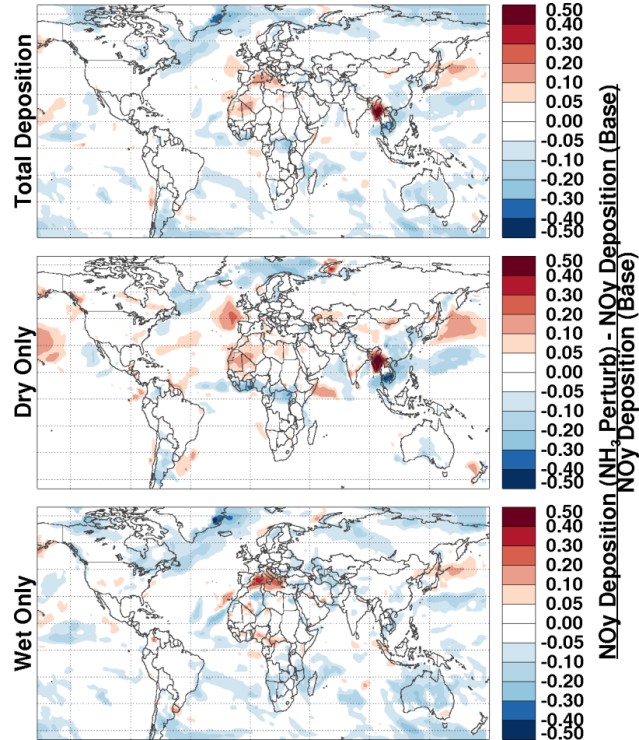

2 Figure 8: Sensitivity of simulated NO$_y$ deposition to a 25% perturbation in ammonia

3 emissions in all grid boxes (shown separately for total deposition, dry deposition, and wet

4 deposition).