# Peer review of "Global deposition of total reactive nitrogen oxides from 1996 to 2014 constrained with satellite observations of NO2 columns"

_Atmospheric Chemistry and Physics, 2016_

## Referee Comment (RC1) · Anonymous Referee #1 · 7 Mar 2017

Better quantifying the trends of atmospheric nitrogen deposition is of great importance to understand and to predict its effects on environment. This manuscript has investigated global deposition of oxidized nitrogen (NOy) using long-term (1996-2014) model simulations with emissions constrained by satellite NO2 column observations. It presents a novel approach to estimate the long-term trends in global NOy deposition by using satellite observations as nitrogen input constraints and a chemical transport model to account for physical/chemical processes in the atmosphere. The model results show different deposition trends from 1996 to 2014 depending on regions, which are evaluated with available surface measurements of wet deposition flux. In

particular, the manuscript points out a recent decline of NOy deposition over China since 2011.

The study is overall well conducted and fits the scope of ACP. The manuscript is well organized and presented. I recommend publish on ACP after the following comments been addressed.

**Specific Comments**
1) Page 5, Methods:
My first comment is that the present manuscript is missing some description and explanation of the top-down NOx emissions. First, measurements from three satellite instruments are used to estimate the long-term NOx emissions. Are there any instrumental differences among the three satellite products? If so, how do you reconcile them? Second, the three satellite instruments have some overlapping time periods (e.g. 2007-2011 for SCIAMACY and GOME-2). How do you use them for emission estimates during these periods? While these may have been presented in another study, a brief description here will help readers to better understand the method.

And third, I strongly suggest present more information and analyses on the global top-down NOx emissions and their trends during the focused period by adding more text and a figure. This is not clear at present, and will be really helpful to understand the trends in NOy deposition as presented in Figure 6 and 7.

2) Page 6, Line 15:
"Monthly mean simulated NO2 columns are calculated using days with coincident satellite observations". How do you select coincident days when using monthly-mean gridded satellite NO2 observations (Line 11)? And how do you sample the model simulation? Please clarify.

3) Page 6, Line 11-17:
Do you mean you do not change the seasonality of NOx emissions in the model? Please clarify. And what is the NOx emission seasonality in the model? This is not described in the Appendix.

4) Page 17, Line 1:
Is there any trends in the export efficiency or changes in the export fraction during the period 1996-2014 over the US and Asia? From Line 11 below, it appears that the export fractions over Europe have a decreasing trend.

5) Page 17, Line 22-24:
A recent study on atmospheric nitrogen deposition over China reported a NOy export fraction of 36% (Zhao et al., 2017), not that different from the values for Europe and the US, compared with 24% in this study. Can you explain why? different NOx emissions, inclusion of adjacent oceans, or model horizontal resolution?

Reference: Zhao, Y., Zhang, L., Chen, Y., Liu, X., Xu, W., Pan, Y., and Duan, L.: Atmospheric nitrogen deposition to China: A model analysis on nitrogen budget and critical load exceedance, Atmospheric Environment, 153, 32-40, 2017.

6) Page 18, Line 30:
Please explain "perturbing NH3 emissions everywhere". Increase or decrease? Do you change all anthropogenic and natural NH3 emissions, including the oceanic NH3 emissions?

7) Page 20, Line 23, 25, 27
The unit "kg N ha-1 yr-2" here might be confusing. Suggest add here "at a rate of . . ." or use annual deposition changes during the period.

[Figure]

8) Page 22, Appendix, Line 27-29:
Energy statistics are used to scale emissions between 1996 and 2010. How about emissions after 2010? Please clarify.

9) Page 44, Figure 8:
Please state in the figure caption that the sensitivity test is for the year 2012.

---

## Referee Comment (RC2) · Anonymous Referee #3 · 8 Apr 2017

Overall, this is an excellent paper that estimates the trends in deposition of NOy using satellite observations of NO2 as constraint.

My major comment is that I recommend a coherent section on model and data uncertainties that may affect your analysis and conclusions. Here are some examples of what such a discussion may include:

Appendix 1

MERRA meteorological fields: Are there any biases in precipitation or transport that may affect your results, such as through simulated wet deposition? Are there any

known biases that change over time in MERRA, such as occur as new observations are brought into the assimilation system over your 20 year simulation period? These are important biases to discuss as global coverage of surface observations (e.g., wet deposition) are sparse over most of the globe.

GEOS-Chem: No model is perfect? Any known issues?

Chemistry: What are the known chemistry uncertainties in the relevant reaction mechanisms? You've answered this with your sensitivity test in Section 3.4.

Emissions: Are there biases? For instance, are the NEI NOx emissions biased?

Travis, K. R. et al., 2016. 'Why do Models Overestimate Surface Ozone in the Southeastern United States?', Atmospheric Chemistry & Physics, 16, 13561-13577, doi:10.5194/acp-16-13561-2016,2016.

Section 3.4: How does the model simulation of ammonia compare to observations, such as from AIRS, and the very long record of SO2, such as from the same instruments that you use for NO2?

Warner, J. X., Wei, Z., Strow, L. L., Dickerson, R. R., and Nowak, J. B.: The global tropospheric ammonia distribution as seen in the 13-year AIRS measurement record, Atmos. Chem. Phys., 16, 5467-5479, doi:10.5194/acp-16-5467-2016, 2016.

Section 2: Satellite NO2: It is no easy task to create an inter-consistent long-term data record using multiple satellite observations, so this topic deserves some discussion. What are the uncertainties and potential biases? For example, a priori vertical profiles change over time.

My minor concerns are:

Page 4, Line 9: Since the topic of this Nowlan paper is similar and from the same group, it may be worth a sentence describing the major conclusion of this paper and how your manuscript is different/better. In fact, you may want to do briefly so the same

for the other papers mentioned in this same paragraph.

Figure 2: The two rows of plots look identical. Is there any way to show differences between the two periods? If not, I'm not sure it's helpful to show both rows.

---

## Author Response (AR1)

Re: Manuscript acp-2016-1100, "Global deposition of total reactive nitrogen oxides from 1996 to 2014 constrained with satellite observations of NO2 columns" by Jeffrey A. Geddes and Randall V. Martin

Dear Handling Editor,

Please find attached our full response to the reviewer comments. We respond to each reviewer comment individually, followed by the relevant changes to our manuscript.

A full copy of our manuscript with tracked changes is included.

Thank you for your time and consideration of our manuscript.

Jeffrey A. Geddes jgeddes@bu.edu

We thank the reviewer very much for their constructive comments. We respond to each comment individually below, followed by changes to the manuscript.

**Reviewer Comment:** My first comment is that the present manuscript is missing some description and explanation of the top-down NOx emissions. First, measurements from three satellite instruments are used to estimate the long-term NOx emissions. Are there any instrumental differences among the three satellite products? If so, how do you reconcile them?

**Author Response:** The referee makes a good point bringing up the instrumental differences between satellite products (a similar concern was raised by another reviewer). The most important differences are the overpass times and the horizontal footprint of individual observations. First, the GOME, SCIAMACHY, and GOME-2 overpass times are roughly 10:30 a.m., 10:00 a.m., and 9:30 a.m. respectively. Second, their horizontal footprints are roughly 320 km x 40 km, 60 km x 30 km, and 80 km x 40 km respectively. We reconciled these three records by consulting published literature and by closely examining the overlapping time periods ourselves, and concluded that a consistent time series is achieved without requiring additional corrections. The reason for this is largely because the daily satellite observations were all gridded to a regular coarse grid of 2° x 2.5° latitude by longitude. Using a comparison with long-term ground-based MAX-DOAS observations, Irie et al. (2012) demonstrated that there is no inherent biases in either SCIAMACHY or GOME-2 that would preclude their combination into a single record. The work of van der A (2008) and Konovalov et al. (2010) show that a self-consistent record can be achieved by downgrading the spatial footprint of the higher resolving instruments (e.g. through smoothing or convolution) to that of the lowest resolving instrument. This is what we have achieved by gridding all the observations to 2° x 2.5°. The combination of these observations is also aided by the fact that the retrieval algorithm for obtaining tropospheric $NO_2$ column density from all three instruments is the same (http://www.temis.nl/airpollution/no2.html).

We further examined our approach by inspecting the timeseries from individual 2° x 2.5° pixels over selected populated regions. These are shown in Figure D1.

Given the evidence from this figure, and consensus in the literature, we concluded that the instrumental differences between instruments are inconsequential to our analysis.

In response to the referee's comment, we have modified our manuscript to include the additional citations and to elaborate on our reasoning for combining the satellite instrument records despite their instrumental differences:

"We calculate top-down surface NOx emissions from 1996 to 2014 using observations from GOME (1995-2003), SCIAMACHY (2002-2011) and GOME-2 (2007- ). The similar overpass time of these three instruments (from about 9:30 a.m. to 10:30 a.m. local time) facilitates their combination to provide consistent long-term coverage (van der A et al., 2008; Konovalov et al. 2010; Geddes et al., 2016; Hilboll et al., 2013). We achieve consistency across all three instruments despite their varying pixel sizes (320 km x 40 km, 60 km x 30 km, and 80 km x 40 km for GOME, SCIAMACHY, and GOME-2 respectively) by gridding the daily observations from each to a regular coarse grid of 2° x 2.5° latitude by longitude."

[Figure]

Figure D1: Monthly mean tropospheric NO2 from GOME, SCIAMACHY, and GOME-2 showing consistent agreement during overlap between instruments.

**RC:** Second, the three satellite instruments have some overlapping time periods (e.g. 2007-2011 for SCIAMACY and GOME-2). How do you use them for emission estimates during these periods? While these may have been presented in another study, a brief description here will help readers to better understand the method.

**AR:** In the case of GOME and SCIAMACHY, their period of overlap is small (June 2002 – April 2003). We use GOME observations alone for 2002, and SCIAMACHY observations alone for 2003. During the overlap between SCIAMACHY and GOME-2 (2007-2011), we use the GOME-2 observations given its more frequent global coverage (roughly every day) compared to SCIAMACHY (roughly every 6 days). We therefore use SCIAMACHY from years 2003-2006, and GOME-2 from years 2007-2014.

In response to the referee's comment, we have added the following to our manuscript:

"In our study, we use GOME observations for years 1996 to 2002, SCIAMACHY observations for years 2003 to 2006, and GOME-2 observations for years 2007 to 2014".

**RC:** And third, I strongly suggest present more information and analyses on the global top-down NOx emissions and their trends during the focused period by adding more text and a figure. This is not clear at present, and will be really helpful to understand the trends in NOy deposition as presented in Figure 6 and 7.

**AR:** We thank the referee for their suggestion, and agree that including details on the top-down emissions would be useful. We believe a table would be the optimal way to share this information. We have also included anthropogenic $HNO_3$ fluxes (from shipping emissions) to our calculation of total emissions to demonstrate full mass balance with $NO_y$ deposition.

In response to this comment, we have added the following material to our manuscript:

"Table 1 shows the annual global top-down NOx emissions from our calculations. We derive global mean satellite-constrained NOx emissions from 1996-2014 of $55.6 \pm 3.4$ Tg N yr$^{-1}$. Our top-down global NOx emissions for 2001 of 52.3 Tg N are consistent with the mean from over 20 models used in the Coordinated Model Studies Activities of the Task Force on Hemispheric Transport of Air Pollution (HTAP) for the same year of $46.6 \pm 7.8$ Tg N (Vet et al. 2014)."

Table 1: Global top-down NOx emissions calculated using the finite mass balance inversion approach in combination with observations from GOME, SCIAMACHY, and GOME-2.

| Year | Global NOx Emissions (Tg N yr$^{-1}$)[a] |
|---|---|
| 1996 | 60.1 |
| 1997 | 58.4 |
| 1998 | 59.2 |
| 1999 | 59.6 |
| 2000 | 53.4 |
| 2001 | 52.3 |
| 2002 | 55.1 |
| 2003 | 50.1 |
| 2004 | 51.5 |
| 2005 | 51.2 |
| 2006 | 50.0 |
| 2007 | 54.7 |
| 2008 | 56.1 |
| 2009 | 55.9 |
| 2010 | 57.5 |
| 2011 | 58.9 |
| 2012 | 59.3 |
| 2013 | 58.5 |
| 2014 | 54.0 |
| Mean | $55.6 \pm 3.4$ |

[a] Includes anthropogenic $HNO_3$ flux of $2.3 \pm 0.1$ Tg N yr$^{-1}$.

**RC:** Monthly mean simulated NO2 columns are calculated using days with coincident satellite observations". How do you select coincident days when using monthly-mean gridded satellite NO2 observations (Line 11)? And how do you sample the model simulation? Please clarify.

**AR:** First, we grid the daily NO2 tropospheric column retrievals to a regular 2° x 2.5° grid, then calculate monthly means ourselves. Therefore, we are able to keep track of which 2° x 2.5° grid boxes have satellite observations that pass quality control on each individual day. From our simulations we output late morning mean vertical NO2 column profiles every day. Using this information, we calculate monthly mean tropospheric NO2 columns from the model simulation only on days that are coincidently sampled by successful satellite observations.

In response to the referee's comment, we have clarified this in our manuscript in the following places:

"We achieve consistency across all three instruments despite their varying pixel sizes (320 km x 40 km, 60 km x 30 km, and 80 km x 40 km for GOME, SCIAMACHY, and GOME-2 respectively) by gridding the daily observations from each to a regular coarse grid of 2° x 2.5° latitude by longitude."

"In all cases, monthly mean simulated NO2 columns are calculated using days with coincident satellite observations. The simulated NO2 vertical column is output daily for late morning."

**RC:** Do you mean you do not change the seasonality of NOx emissions in the model? Please clarify. And what is the NOx emission seasonality in the model? This is not described in the Appendix

**AR:** The referee is correct. We have not changed the seasonality of NOx emissions in the model. We assume that the scaling factor determined from coincidently sampled model and satellite NO2 tropospheric columns applies uniformly to the model prior emissions all year long regardless of whether successful satellite observations are available.

In response to the referee's comment, we have modified our manuscript to clarify our approach:

"In all cases, monthly mean simulated NO2 columns are calculated using days with coincident satellite observations. The simulated NO2 vertical column is output daily for late morning. We calculate scaling factors for every month with available satellite observations, then calculate an annual mean scaling factor that is used to infer annual mean from the mean monthly top-down emissions. Our top-down emissions retain the same seasonality as the prior emissions to mitigate concerns about seasonally missing data (such as from snow or monsoonal clouds)."

We have also added the following details regarding the emission seasonality to the Appendix:

"Monthly scaling of NOx emissions are included in North America (based on the VISTAS inventory), Europe (based on the EMEP inventory), and Asia (based on the Zhang et al. (2009) inventory). Monthly scaling of EDGAR emissions is based on the seasonality from the Global Emission Inventory Activity (Benkovitz et al 1996)."

**RC:** Is there any trends in the export efficiency or changes in the export fraction during the period 1996-2014 over the US and Asia? From Line 11 below, it appears that the export fractions over Europe have a decreasing trend.

**AR:** We thank the reviewer for their comment, and have evaluated the statistical significance of trends in export fraction over each region in the same manner as our evaluation of long term trends in deposition.

In response to the referee's comment, we have added the following details to our manuscript:

"We estimate a similar fraction of $NO_x$ export from the continental US using our observationally-constrained simulation (34% $\pm$ 2% from 1996-2014), with a small declining trend from a maximum of 38% in 1999 to a minimum of 31% in 2013.

"We calculate mean export of $NO_x$ emissions from western European countries to be 45% $\pm$ 4%, with a notable decreasing trend from a maximum of 50% in 1997 to a minimum of 39% in 2014."

"We estimate that an average of 24% $\pm$ 4% emissions from China are exported, varying over time from as little as 15% of emissions in 1998 to a maximum of 31% of emissions in 2011 (an overall increasing trend)."

**RC:** A recent study on atmospheric nitrogen deposition over China reported a NOy export fraction of 36% (Zhao et al., 2017), not that different from the values for Europe and the US, compared with 24% in this study. Can you explain why? different NOx emissions, inclusion of adjacent oceans, or model horizontal resolution?

**AR:** We thank the reviewer for this important reference, which we have added to our manuscript. In our opinion, the most obvious explanation for the discrepancy is in model resolution or perhaps the rapidly changing satellite-constrained emissions over this period of time, which peak in China in 2011.

In response to the referee's comment, we have made the following changes to our manuscript:

"We estimate that an average of 24% $\pm$ 4% emissions from China are exported, varying over time from as little as of 15% of emissions in 1998 to a maximum of 31% of emissions in 2011 (an overall increasing trend). Zhao et al. (2017) used a higher resolution (0.5° x 0.667°) GEOS-Chem simulation and estimated that 36% of China's NOx emissions over 2008-2012 are exported. We calculate an export fraction of around 27% for the same time period. The discrepancy between the two estimates may be attributed to the coarser horizontal resolution of our simulation (2° x 2.5°), pointing to important resolution-dependent effects in global simulations of deposition. Other factors may include the use of different NOx emissions (our satellite-constrained emissions indicate rapid change over this period of time), and the treatment of adjacent oceans."

**RC:** Please explain "perturbing NH3 emissions everywhere". Increase or decrease? Do you change all anthropogenic and natural NH3 emissions, including the oceanic NH3 emissions?

**AR:** In response to the referee's comments, we have clarified the approach in our manuscript:

"Contemporary emissions of NH3 are highly uncertain (Reis et al., 2009), so we perform a sensitivity experiment by perturbing (increasing) all anthropogenic and natural NH3 emissions in the model by 25% for the year 2012."

**RC:** The unit "kg N ha$^{-1}$ yr$^{-2}$" here might be confusing. Suggest add here "at a rate of…" or use annual deposition changes during the period.

**AR:** We thank the referee for their suggestion. In response, we have modified our manuscript to use the following wording:

"NOy deposition declined most steeply throughout the northeastern United States at a rate of up to -0.6 kg N ha$^{-1}$ yr$^{-2}$"

"In Europe, statistically significant declines at a rate of up to -0.1 kg N ha$^{-1}$ yr$^{-2}$ are seen over some western countries."

"On the other hand, NO$_y$ deposition has increased substantially throughout East Asia, exceeding a rate of +0.6 kg N ha$^{-1}$ yr$^{-2}$ in some parts."

**RC:** Energy statistics are used to scale emissions between 1996 and 2010.   How about emissions after 2010? Please clarify.

**AR:** We clarify this in our manuscript:

"For other species and for emissions beyond 2010, the closest available year is used."

**RC:** Please state in the figure caption that the sensitivity test is for the year 2012.

**AR:** In response to the reviewer's comment, we have added this to the figure caption.

**REFERENCES:**

Benkovits et al. (1996), Global gridded inventories of anthropogenic emissions of sulfur and nitrogen, *Journal of Geophysical Research-Atmospheres,* 101, 29239–29253

Irie et al. (2012), Quantitative bias estimates for tropospheric NO2 columns retrieved from SCIAMACHY, OMI, and GOME-2 using a common standard for East Asia, *Atmospheric Measurement Techniques*, 5, 2403-2411

Konovalov et al. (2010) Multi-annual changes of NOx emissions in megacity regions: nonlinear trend analysis of satellite measurement based estimates, Atmos. Chem. Phys., 10, 8481–8498

van der A et al. (2008), Trends, seasonal variability, and dominant NOx source derived from a ten year record of NO2 measured from space, *Journal of Geophysical Research-Atmospheres, 113, doi:10.1029/2007JD009021*

Zhao et al. (2017), Atmospheric nitrogen deposition to China: A model analysis on nitrogen budget and critical load exceedance, *Atmospheric Environment,* 153, 32-40

We thank the reviewer very much for their constructive comments. We respond to each comment individually below, followed by changes to the manuscript.

**Reviewer Comment:** My major comment is that I recommend a coherent section on model and data uncertainties that may affect your analysis and conclusions. Here are some examples of what such a discussion may include:

MERRA meteorological fields: Are there any biases in precipitation or transport that may affect your results, such as through simulated wet deposition? Are there any known biases that change over time in MERRA, such as occur as new observations are brought into the assimilation system over your 20 year simulation period? These are important biases to discuss as global coverage of surface observations (e.g., wet deposition) are sparse over most of the globe.

GEOS-Chem: No model is perfect? Any known issues?

Chemistry: What are the known chemistry uncertainties in the relevant reaction mechanisms? You've answered this with your sensitivity test in Section 3.4.

Emissions: Are there biases? For instance, are the NEI NOx emissions biased?

**Author Response:** We agree with the referee that there could be a larger discussion on model and data uncertainties that may affect our analysis.

In response to the reviewer's comment, we have added the following discussion section to our manuscript:

"3.5 Other Considerations

A number of other uncertainties are important in an inversion of satellite NO2 columns to calculate surface NOx emissions and simulate of long-term NOy deposition. These can depend on, for example, the choice of inversion approach, errors in the satellite retrieval, and uncertainties in model processes (e.g. emissions, boundary layer mixing, chemical NOx sinks, meteorology, and dry deposition).

Cooper et al. (2017) found that the finite mass balance inversion approach used here can be improved upon by using an iterative method that performs with similar accuracy as a four-dimensional variational data assimilation. Multi-constituent data assimilation also shows considerable promise for constraints on surface NOx emissions (Myazaki et al. 2017). Satellite retrieval algorithms continue to develop with advances that will improve the accuracy of future estimates of satellite-constrained NOy deposition.

Uncertainties in model processes are also of interest. For example, uncertainties in the chemical sink of NOx alone (e.g. the rate of HNO3 formation, heterogeneous loss of N2O5 onto aerosol) can have a substantial impact on top-down emissions estimates (Stavrakou et al., 2013), suggesting more fundamental work in constraining these processes is required. Lin et al. (2010) found that top-down NOx emissions estimates over East Asia are sensitive to other model uncertainties including planetary boundary layer mixing scheme, lightning emissions, diurnal profile of emissions, and a-priori NOx, CO, and VOC emissions. Uncertainties in model meteorology are also important. For example, the MERRA precipitation fields used in our study are known to correlate weakly with observational datasets (Rienecker et al., 2011), but improvements can be expected from MERRA-2 due to the inclusion of gauge- and satellite-based precipitation corrections (Reichle et al., 2017). Finally, dry deposition schemes are also highly variable among models (Flechard et al., 2011; Hardacre et al., 2015), and future work in dry deposition evaluation should be a priority.

Nonetheless, despite these uncertainties we find a high degree of consistency between observations and our predictions in the long-term changes to deposition. Evidence continues to emerge about potential biases in bottom-up inventories (e.g. Travis et al. 2016), and our observational constraint on NOx emissions mitigates against such biases. We expect continued advancements in inversion approaches, satellite retrieval algorithms, and fundamental atmospheric chemistry processes will allow for increasingly accurate satellite-based constraints on deposition."

**RC:** Section 3.4: How does the model simulation of ammonia compare to observations, such as from AIRS, and the very long record of SO2, such as from the same instruments that you use for NO2

**AR:** The reviewer raises a good question. GEOS-Chem ammonia simulations have been explored by other groups (e.g. Schiferl et al. 2016), and suggest that the model can underestimate interannual variability and concentrations of ammonia. A new NH3 emission inventory is now available (Paulot et al., 2014), but this inventory is still only representative of a short period of time (2005-2008), thus potentially introducing similar errors in other years of our analysis anyway. Our perturbation experiment with NH3 emissions (Section 3.4) is an attempt to investigate the dependence of NOy deposition on NH3 emission uncertainties. We found that reasonable uncertainties/changes in NH3 emissions (~25%) are generally inconsequential to the spatial distribution of predicted NOy deposition over land. We did not feel the need to repeat such an experiment with SO2 emissions, since SO2 emissions tend to be less uncertain than NH3 emissions and will therefore have even less of an impact on the predicted spatial distribution of NOy deposition. We argue that an evaluation of GEOS-Chem simulated NH3 and SO2 is thus beyond the scope of our study, and would not have substantial bearing on the conclusions in our manuscript.

**RC:** It is no easy task to create an inter-consistent long-term data record using multiple satellite observations, so this topic deserves some discussion. What are the uncertainties and potential biases? For example, a priori vertical profiles change over time

**AR:** We agree with the reviewer that creating a long-term record across multiple satellite records is not necessarily a trivial task (a similar concern was raised by another reviewer). In our case, we concluded that a consistent time series can be achieved without requiring additional corrections. The reason for this is largely because the daily satellite observations were all gridded to a regular coarse grid of 2° x 2.5° latitude by longitude. Using a comparison with long-term ground-based MAX-DOAS observations, Irie et al. (2012) demonstrated that there is no inherent biases in either SCIAMACHY or GOME-2 that would preclude their combination into a single record. The work of van der A (2008) and Konovalov et al. (2010) show that a self-consistent record can be achieved by downgrading the spatial footprint of the higher resolving instruments (e.g. through smoothing or convolution) to that of the lowest resolving instrument. This is what we have achieved by gridding all the observations to 2° x 2.5°. The combination of these observations is also aided by the fact that the retrieval algorithm for obtaining tropospheric NO2 column density from all three instruments is the same ([http://www.temis.nl/airpollution/no2.html](http://www.temis.nl/airpollution/no2.html)), and by the fact that their overpass times are similar (between 9:30 a.m. and 10:30 a.m. local time).

We tested our approach by inspecting the timeseries from individual 2° x 2.5° pixels over selected populated regions. These are shown in Figure D1.

[Figure]

Figure D1: Monthly mean tropospheric NO2 from GOME, SCIAMACHY, and GOME-2 show consistent agreement during overlap between instruments.

Given this and the consensus in the literature, we conclude that the instrumental differences between each satellite instrument are inconsequential to our analysis.

In response to the referee's comment, we have modified our manuscript to include additional citations, and to elaborate on our reasoning for combining the satellite instrument records despite their instrumental differences:

"We calculate top-down surface NOx emissions from 1996 to 2014 using observations from GOME (1995-2003), SCIAMACHY (2002-2011) and GOME-2 (2007- ). The similar overpass time of these three instruments (from 9:30 a.m. to 10:30 a.m. local time) facilitates their combination to provide consistent long-term coverage (van der A et al., 2008; Konovalov et al. 2010; Geddes et al., 2016; Hilboll et al., 2013). We achieve consistency across all three instruments despite their varying pixel sizes (320 km x 40 km, 60 km x 30 km, and 80 km x 40 km for GOME, SCIAMACHY, and GOME-2 respectively) by gridding the daily observations from each to a regular coarse grid of 2° x 2.5° latitude by longitude."

The referee also brings up the issue of changing a-priori vertical profiles over time. We agree that vertical profile shapes are an integral component of the tropospheric NO2 column retrieval, and indeed are a significant contributor to column error (at least ~10% according to Boersma et al. (2004), and up to ~100% according to Laughner et al. (2016)). However, we are limited by the current generation of available global tropospheric NO2 retrieval algorithms. Advances in satellite-derived tropospheric NO2 column retrievals will be integral to improvements in top-down emissions estimates. In response to the reviewer comment, we address this in modifications we have made to the manuscript:

"The error in individual satellite-derived tropospheric $NO_2$ column retrievals to be around 35-60% for polluted scenes and greater than 100% for clean regions (Boersma et al. 2004)."

"We expect continued advancements in inversion approaches, satellite retrieval algorithms, and fundamental atmospheric chemistry processes will allow for increasingly accurate satellite-based constraints on deposition."

**RC:** Page 4, Line 9: Since the topic of this Nowlan paper is similar and from the same group, it may be worth a sentence describing the major conclusion of this paper and how your manuscript is different/better. In fact, you may want to do briefly so the same for the other papers mentioned in this same paragraph.

**AC:** We thank the reviewer for this suggestion. We agree that it would be insightful to summarize the important points from Nowlan et al., although the statistical approaches taken by the other examples vary substantially from our approach and are less directly comparable. In response to this comment, we have included the following in our manuscript:

"Nowlan et al. (2014) demonstrated how satellite-inferred surface concentrations of NO2 can be combined with modeling to produce spatially continuous estimates of NO2 dry deposition fluxes. They found that dry deposition of $NO_2$ contributes as much as 85% of total NOy deposition in urban areas, but only represents 3% of global NOx emitted. The remaining 97% of global NOy deposition is made up of both wet and dry deposition of other reactive nitrogen oxide compounds that are not directly observed by satellite-based instruments."

**RC:** Figure 2. The two rows of plots look identical. Is there any way to show the differences between the two periods. If not, I'm not sure it's helpful to show both rows.

**AC:** We are very grateful to the reviewer for their careful attention to this figure. In fact, the two panels are identical and this was an error in the original figure (both top and bottom accidentally show results for 2000-2002). We have corrected this mistake and sincerely apologise for the error. A new version of this figure will be uploaded with the corrected manuscript, and is included here as Figure D2 for the reviewer's reference. The differences between the two time periods are now more obvious. Declines in wet nitrate deposition over this period are evident over North America and parts of Western Europe, while increases are evident over East Asia. Note that the observational coverage is also slightly different across the two periods.

[Figure]

Figure D2: Corrected version of Figure 2 from the manuscript, showing results from the 2000-2002 comparison in the top panels, and results from the 2005-2007 comparison in the bottom panels.

are exported. We calculate an export fraction of around 27% for the same time period. The discrepancy between the two estimates may be attributed to the coarser horizontal resolution of our simulation (2° x 2.5°), pointing to important resolution-dependent effects in global simulations of deposition. Other factors may include the use of different NOx emissions (our satellite-constrained emissions indicate rapid change over this period of time), and the treatment of adjacent oceans.

[revised manuscript text omitted]

### 3.5  Other Considerations

A number of other uncertainties are important in an inversion of satellite $NO_2$ columns to calculate surface $NO_x$ emissions and simulate of long-term $NO_y$ deposition. These can depend on, for example, the choice of inversion approach, errors in the satellite retrieval, and uncertainties in model processes (e.g. emissions, boundary layer mixing, chemical $NO_x$ sinks, meteorology, and dry deposition).

Cooper et al. (2017) found that the finite mass balance inversion approach used here can be improved upon by using an iterative method that performs with similar accuracy as a four- dimensional variational data assimilation. Multi-constituent data assimilation also shows considerable promise for constraints on surface $NO_x$ emissions (Myazaki et al. 2017). Satellite retrieval algorithms continue to develop with advances that will improve the accuracy of future estimates of satellite-constrained $NO_y$ deposition.

Uncertainties in model processes are also of interest. For example, uncertainties in the chemical sink of $NO_x$ alone (e.g. the rate of $HNO_3$ formation, heterogeneous loss of $N_2O_5$

onto aerosol) can have a substantial impact on top-down emissions estimates (Stavrakou et al., 2013), suggesting more fundamental work in constraining these processes is required. Lin et al. (2010) found that top-down $NO_x$ emissions estimates over East Asia are sensitive to other model uncertainties including planetary boundary layer mixing scheme, lightning emissions, diurnal profile of emissions, and a-priori NOx, CO, and VOC emissions.

Uncertainties in model meteorology are also important. For example, the MERRA

precipitation fields used in our study are known to correlate weakly with observational datasets (Rienecker et al., 2011), but improvements can be expected from MERRA-2 due to the inclusion of gauge- and satellite-based precipitation corrections (Reichle et al., 2017).

Finally, dry deposition schemes are also highly variable among models (Flechard et al., 2011;

Hardacre et al., 2015), and future work in dry deposition evaluation should be a priority.

Nonetheless, despite these uncertainties we find a high degree of consistency between observations and our predictions in the long-term changes to deposition. Evidence continues to emerge about potential biases in bottom-up inventories (e.g. Travis et al. 2016), and our observational constraint on NOx emissions mitigates against such biases. We expect continued advancements in inversion approaches, satellite retrieval algorithms, and fundamental atmospheric chemistry processes will allow for increasingly accurate satellite-based constraints on deposition.

[revised manuscript text omitted]

[a] Includes anthropogenic HNO$_3$ flux of 2.3 ± 0.1 Tg N yr$^{-1}$.

[Figure]

Figure 1: Long-term (1996-2014) mean $NO_y$ deposition derived from the GEOS-Chem simulation constrained by satellite observations of $NO_2$ columns from the GOME, SCIAMACHY, and GOME-2 instruments (top). Mean ratio of simulated dry $NO_y$ deposition to total $NO_y$ deposition (bottom).

[Figure]

Figure 2: Annual wet NO₃- deposition from measurements available through the World Data Centre for Precipitation Chemistry, and from the GEOS-Chem simulation constrained with satellite observations of NO$_2$. Two time periods are represented: 2000-2002 and 2005-2007.

[Figure]

Figure 3: Scatter plot of the satellite-constrained simulated wet $NO_3^-$ deposition vs. measurements available through the World Data Centre for Precipitation Chemistry for specific subsets of the data. The red lines show the result of a reduced major axis linear regression. In the right column, the blue line shows the fit across all data and the red line shows the fit excluding the two circled data points that are discussed in the text (reported statistics refer to the red line fit).

[Figure]

Figure 4: Long-term (1996-2014) wet $NO_3^-$ deposition from available regional network measurements (top: NADP and CAPMON; middle: EMEP; bottom: EANet), and from the GEOS-Chem simulation constrained with satellite observations of $NO_2$.

[Figure]

Figure 5. Long-term trend (1996-2014) in the satellite-constrained simulation of $NO_y$ deposition. (A) Annual mean; (B) December-January-February; (C) March-April-May; (D) June-July-August; (E) September-October-November. Diagonal hatching represents trend significance ($p < 0.01$). Hatching from top-left to bottom-right indicates a decreasing trend; hatching from bottom-left to top-right indicates an increasing trend.

[Figure]

Figure 6: Timeseries of annually integrated dry and wet NOy deposition over specific regions (North America, Europe, and East Asia) as defined by the dashed rectangles in Figure 5.

[Figure]

Figure 7: Long-term (1996-2014) trends in wet $NO_3^-$ deposition from available regional network measurements (as in Figure 4), and from the GEOS-Chem simulation constrained by satellite observations of $NO_2$. Closed circles around the measurements indicate significant trends ($p < 0.01$); hatching indicates statistical significance in the simulation.

[Figure]

Figure 8: Sensitivity of simulated NO$_y$ deposition in 2012 to a 25% perturbation (increase) in ammonia emissions in all grid boxes (shown separately for total deposition, dry deposition, and wet deposition).